# Reversible hydrogen spillover in Ru-WO$_{3-x}$ enhances hydrogen evolution activity in neutral pH water splitting

Jiadong Chen [1,2,5], Chunhong Chen[1,5], Minkai Qin[1,5], Ben Li[1], Binbin Lin[1], Qing Mao[3], Hongbin Yang[2], Bin Liu [2,4] ✉ & Yong Wang [1] ✉

Noble metal electrocatalysts (e.g., Pt, Ru, etc.) suffer from sluggish kinetics of water dissociation for the electrochemical reduction of water to molecular hydrogen in alkaline and neutral pH environments. Herein, we found that an integration of Ru nanoparticles (NPs) on oxygen-deficient WO$_{3-x}$ manifested a 24.0-fold increase in hydrogen evolution reaction (HER) activity compared with commercial Ru/C electrocatalyst in neutral electrolyte. Oxygen-deficient WO$_{3-x}$ is shown to possess large capacity for storing protons, which could be transferred to the Ru NPs under cathodic potential. This significantly increases the hydrogen coverage on the surface of Ru NPs in HER and thus changes the rate-determining step of HER on Ru from water dissociation to hydrogen recombination.

Hydrogen, with high gravimetric energy density, is an ideal candidate to replace the traditional fossil fuels and also a pivotal ingredient for essential industrial chemicals (e.g., petroleum refining and ammonia synthesis)[1–5]. Water electrolysis driven by renewable electricity offers great promises for eco-friendly hydrogen production. Electrolysis of water can be realized in acidic, neutral and alkaline environment, among which, water reduction in neutral/alkaline medium is much more sluggish because of the slow water dissociation reaction[6–8]. Consequently, even platinum (Pt), the state-of-the-art hydrogen evolution reaction (HER) catalyst, shows two to three orders of magnitude lower activity in neutral/alkaline medium as compared to acidic medium[9]. Although HER in acidic condition exhibits better activity, equipment and catalyst corrosion limit the lifetime of operation. Neutral media provides a more favorable condition for catalysts to remain stable and less corrosive environment for electrolysers[10]. And electrolysers capable of operating in neutral media offer the possibility of achieving hydrogen production directly from seawater without the need for desalination[10,11].

In practical application, catalysts usually operate at large overpotentials to achieve large current densities. In this case, early studies show that HER in neutral/alkaline medium starts from water dissociation ($M + e^- + H_2O \rightarrow M - H_{ad} + OH^-$, where M refers to the active site and $H_{ad}$ stands for adsorbed H), followed by either the Heyrovsky reaction ($H_2O + e^- + M - H_{ad} \rightarrow M + OH^- + H_2$) or the Tafel reaction ($H_{ad} + H_{ad} \rightarrow H_2$)[10,12,13]. Compared with HER in acidic environment, the additional splitting of water molecules to supply protons in neutral/alkaline medium is more sluggish in kinetics, resulting in low hydrogen coverage ($M - H_{ad}$) on the surface of the catalyst during HER.

To enhance HER in neutral/alkaline medium, prior approaches were mainly dedicated to facilitating sluggish water dissociation reaction by means of incorporating a specific component (e.g., transition metal hydroxide) onto the catalytically active species (e.g., Pt) or inducing surface reconstruction to expose more active sites[14–18]. The water dissociation process releases H$^+$, which will be bound on the surface of the catalyst, undergoing hydrogen recombination to evolve molecular hydrogen. The sluggish water dissociation reaction in neutral/alkaline medium results in a low H coverage on the surface of HER catalyst, which impedes HER catalysis[19–21]. Hydrogen spillover, the migration of activated hydrogen atoms generated by the dissociation of di-hydrogen adsorbed on

[1]Advanced Materials and Catalysis Group, Center of Chemistry for Frontier Technologies, State Key Laboratory of Clean Energy Utilization, Institute of Catalysis, Department of Chemistry, Zhejiang University, Hangzhou 310028, PR China. [2]School of Chemical and Biomedical Engineering, Nanyang Technological University, Singapore 637459, Singapore. [3]School of Chemical Engineering, Dalian University of Technology, Dalian 116024 Liaoning, PR China. [4]Division of Chemistry and Biological Chemistry, School of Physical and Mathematical Sciences, Nanyang Technological University, Singapore 637371, Singapore. [5]These authors contributed equally: Jiadong Chen, Chunhong Chen, Minkai Qin. ✉e-mail: liubin@ntu.edu.sg; chemwy@zju.edu.cn

a metal surface onto a reducible metal oxide support, is a common phenomenon in heterogeneous catalysis[22–26]. Recently, hydrogen spillover strategy has been taken into account for the catalysts design to achieve the compelling HER performance, such as Pt alloys-CoP[27], Pt/CoP[28], and Pt/TiO$_2$[29] electrocatalysts, delivering the optimal HER activity. However, spillover strategies have rarely been studied on neutral HER and the exact mechanism of hydrogen spillover to improve HER is still unclear.

In this work, we develop an effective strategy to significantly increase the H coverage on the catalyst during HER in neutral environment. Specifically, we propose a Ru nanoparticles (NPs) on oxygen-deficient tungsten oxide (Ru-WO$_{3-x}$) system, in which protons inserted into WO$_{3-x}$ can be transferred to Ru NPs during HER, thereby greatly increasing the hydrogen coverage on Ru NPs and enhancing the HER performance. Through combined in situ Raman spectroscopy investigations, electrochemical measurements and DFT calculations, the hydrogen spillover from WO$_{3-x}$ to Ru NPs during HER has been explicitly demonstrated. Consequently, the HER activity of Ru-WO$_{3-x}$ is enhanced by a factor of 24.0 as compared with the commercial Ru/C (5.0 wt.%) electrocatalyst in 1.0 M phosphate buffer solution (PBS) electrolyte.

## Results

### Origin of unsatisfied HER activity of Ru/C in neutral medium

Ru, having a lower cost but comparable hydrogen binding energy as compared to Pt[30–34], is regarded as one of the good candidates to replace Pt in HER. Early studies demonstrated good HER activity of commercial Ru/C in acidic medium[35,36]. However, the HER activity of Ru/C significantly reduced in neutral environment. To figure out the sluggish kinetics of HER on Ru/C in neutral medium, we performed microkinetic analysis. The HER activity of commercial Ru/C (5.0 wt.%) electrocatalyst was first evaluated in 1.0 M PBS (Fig. 1a). The Ru/C electrocatalyst displays a large overpotential of 86 mV at a current density of 10 mA cm$^{-2}$ with a Tafel slope as large as 78 mV dec$^{-1}$. By fitting the electrochemical data using a microkinetic model, the HER on Ru/C in neutral medium was found to be rate-limited at the water dissociation step, which led to the low hydrogen coverage ($\theta_H$) on the surface of Ru during HER (Fig. 1b, Supplementary Note 1, Supplementary Tables 1–6, and Supplementary Figs. 1 and 2). The low $\theta_H$ on Ru/C in HER was experimentally verified by in situ Raman spectroscopy, where no observable peaks appear in the Raman frequency range of Ru-H vibration (Fig. 1c)[37,38]. Therefore, effective strategies need to be proposed to enhance the HER activity of Ru/C in neutral media.

Oxygen-deficient tungsten oxide (WO$_{3-x}$) displays the excellent capability of storing protons in water[39–41]. If the inserted protons in WO$_{3-x}$ are mobile, WO$_{3-x}$ may be used as a proton reservoir to supply H to increase $\theta_H$ on Ru during HER, which thus shall promote the HER

kinetics of Ru in neutral medium. Inspired by this possibility, we prepared Ru NPs on oxygen-deficient WO$_{3-x}$ (Ru-WO$_{3-x}$) electrocatalysts and studied their HER catalysis.

### Synthesis and characterization of Ru-WO$_{3-x}$

Ru-WO$_{3-x}$ was prepared by a three-step method as schematically illustrated in Fig. 2a. WO$_3$/CP was first synthesized by a simple hydrothermal method and then impregnated in RuCl$_3$ solution, followed by a heat treatment in H$_2$/Ar mixed atmosphere (10/90 molar ratio) to form Ru-WO$_{3-x}$/CP. The crystal structure of the as-prepared Ru-WO$_{3-x}$/CP catalyst was examined by X-ray diffraction (XRD) as shown in Fig. 2b. The Ru-WO$_{3-x}$/CP displays clear diffraction peaks of hexagonal WO$_3$ (JCPDS No. 85−2460)[42], but no diffraction peaks related to Ru NPs, possibly due to their small sizes and low content. In addition, the peaks of Ru-WO$_{3-x}$/CP are shifted to high angles, indicating lattice shrinkage, which may be caused by oxygen vacancies[40]. Scanning electron microscopy (SEM) and transmission electron microscopy (TEM) measurements were performed to probe the morphological information. The WO$_3$/CP exhibits nanowires with smooth surfaces grown on carbon fibers (Supplementary Fig. 3a–d). High-resolution TEM (HRTEM) image of a single nanowire gives a lattice spacing of 3.84 Å (Supplementary Fig. 3e)[43,44], which is attributed to the (002) facet of hexagonal WO$_3$ (JCPDS No. 85-2460). Energy dispersive X-ray spectroscopy (EDX) elemental mapping of a single WO$_3$ nanowire reveals uniform distribution of the W and O elements (Supplementary Fig. 3f–h). Comparatively, the Ru-WO$_{3-x}$/CP displays rough nanowires (Supplementary Fig. 4a, b), indicating successful loading of Ru NPs, which is further verified by the TEM measurements (Supplementary Fig. 4c–e). Both lattice spacings resulting from hexagonal WO$_3$ and hexagonal Ru are clearly visible in the HRTEM images of Ru-WO$_{3-x}$/CP (Fig. 2c, d and Supplementary Fig. 4f). On the other hand, the lattice fringes of WO$_3$ at the edge become blurred, which may be due to the formation of oxygen vacancies induced by hydrogen reduction[45]. EDX elemental mappings reveal uniform distribution of Ru, W and O elements (Supplementary Fig. 4g). Moreover, the average size of the Ru NPs on Ru-WO$_{3-x}$/CP is 3.5 nm (Supplementary Fig. 4h). The mass content of Ru in Ru-WO$_{3-x}$/CP was determined to be 5.1 wt.% by inductively coupled plasma optical emission spectrometry (ICP-OES) (Supplementary Table 7). The formation of oxygen vacancies in Ru-WO$_{3-x}$/CP was confirmed by the electron paramagnetic resonance (EPR) measurement as shown in Fig. 2e. To probe the valence states and further explore the oxygen vacancies in Ru-WO$_{3-x}$/CP, X-ray photoelectron spectroscopy (XPS) was performed. Ru $3d_{5/2}$ core level XPS spectrum displays two peaks at 280.28 eV and 281.28 eV, respectively, matching well with Ru(0) and Ru(IV) (Fig. 2f)[30,31]. The Ru $3p$ XPS spectrum shows two pairs of peaks, in which the dominant peaks at 461.88 eV and 484.14 eV can be assigned to Ru $3p_{3/2}$ and Ru $3p_{1/2}$ of Ru(0) and the rest of the peaks are from Ru(IV), manifesting that the Ru precursor has

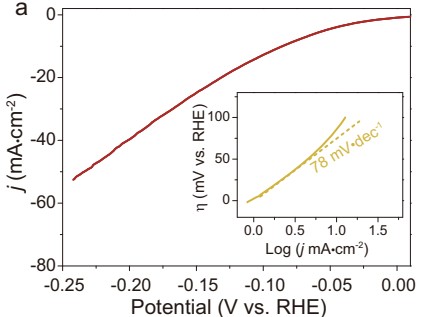
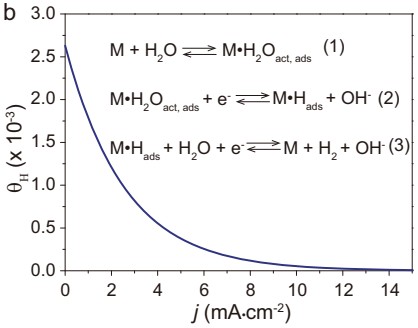
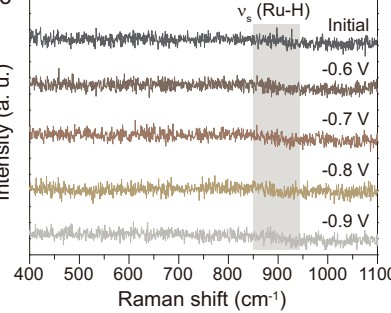

**Fig. 1 | HER performance of commercial Ru/C (5 wt%) in 1.0 M PBS. a** LSV curve and the corresponding Tafel plot for commercial Ru/C (5.0 wt.%) loaded on CP. Scan rate: 2 mV/s. **b** Hydrogen coverage as a function of current density obtained from the microkinetic analysis. The microkinetic model was built based on three elementary reactions. **c** In situ Raman spectra recorded on Ru/C (5.0 wt.%) in 1.0 M PBS in the potential range from −0.6 to −0.9 V vs. Ag/AgCl.

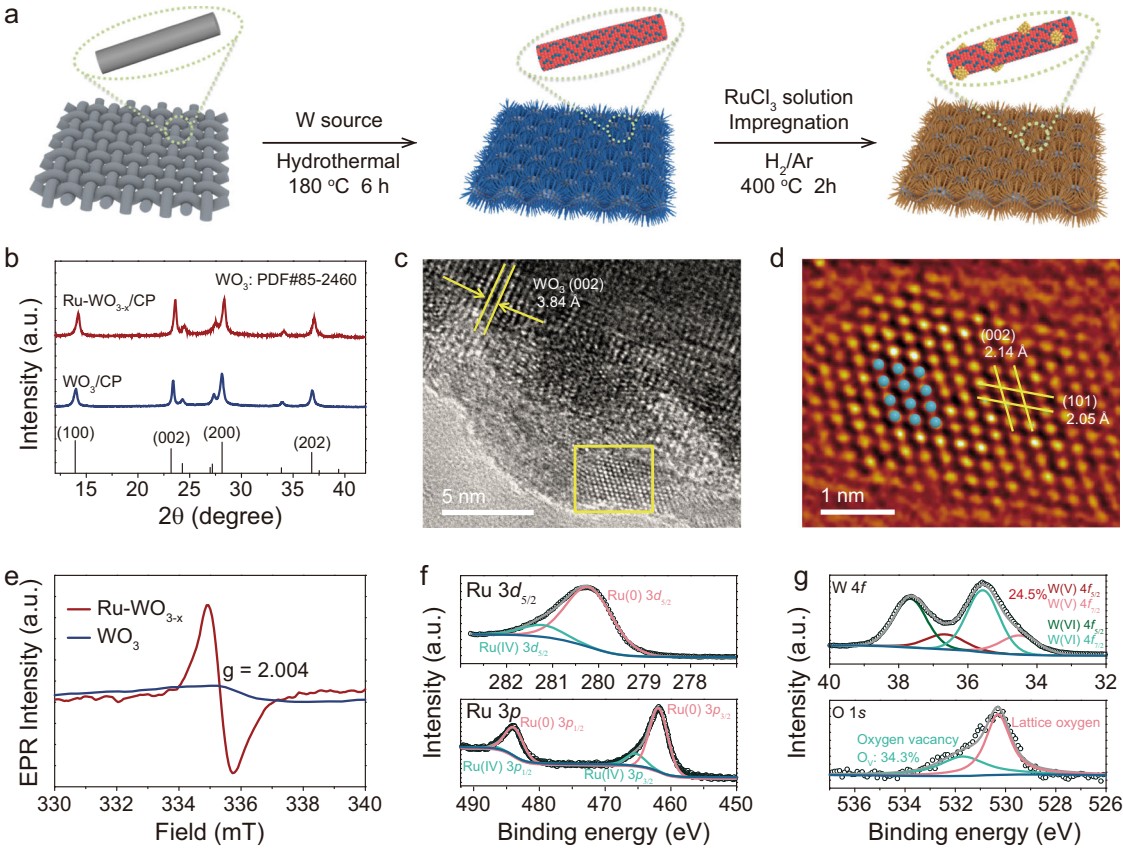

**Fig. 2 | Preparation and characterization of Ru-WO₃₋ₓ/CP. a** Schematic illustration showing the procedure to prepare Ru-WO₃₋ₓ/CP. **b** XRD patterns of WO₃/CP and Ru-WO₃₋ₓ/CP. **c** HRTEM image of Ru-WO₃₋ₓ/CP. **d** Filtered HRTEM image (using ASBF filter) and the corresponding structural model of Ru nanoparticle marked in **c** of Ru-WO₃₋ₓ/CP. The blue spheres represent Ru atoms. **e** EPR spectra of WO₃ and Ru-WO₃₋ₓ. **f** Ru $3d_{5/2}$ and Ru $3p$ XPS spectra of Ru-WO₃₋ₓ/CP. **g** W $4f$ and O 1$s$ XPS spectra of Ru-WO₃₋ₓ/CP.

been successfully reduced to metallic Ru. The W $4f$ XPS spectrum of Ru-WO₃₋ₓ/CP displays two peaks at 35.58 eV and 37.68 eV, which correspond well to the W $4f_{2/7}$ and W $4f_{2/5}$ of W(VI), respectively (Fig. 2g)[40,41]. Additionally, two more deconvoluted peaks at 34.53 eV and 36.63 eV can be assigned to W(V)[43,45]. Notably, a few hydrogen atoms may be induced in the surface of WO₃₋ₓ support in the H₂/Ar reduction process and this may also lead to the appearance of W(V). In O 1$s$ XPS spectrum, two deconvoluted peaks are observed. The peak centered at 531.70 eV is assigned to the OH groups or a lattice oxygen bounded to a W(V) atom (close to a vacancy)[45]. And another peak at 530.30 eV is ascribed to the lattice oxygen. The percentage of oxygen vacancies in Ru-WO₃₋ₓ/CP is determined to be around 34.3%, consistent with the W(V) content deduced from W $4f$ XPS spectrum. In addition, WO₃₋ₓ/CP was prepared, and the corresponding W $4f$ and O1$s$ XPS of WO₃₋ₓ/CP showed that there were also a large number of oxygen vacancies (Supplementary Fig. 5). To further investigate oxygen vacancies in Ru-WO₃₋ₓ and WO₃₋ₓ, we conducted O₂-temperature programmed desorption (O₂-TPD). The corresponding result reveals oxygen vacancies in WO₃₋ₓ and Ru-WO₃₋ₓ, which matches well with the EPR and O 1$s$ XPS results (Supplementary Fig. 6).

## HER performance of Ru-WO₃₋ₓ/CP in neutral media

The HER performance of the as-prepared electrocatalysts was examined in a three-electrode system in N₂ saturated 1.0 M PBS. In comparison to WO₃₋ₓ/CP and commercial Ru/C (5.0 wt.%)/CP, the Ru-WO₃₋ₓ/CP exhibits a greatly improved HER activity, reaching a current density of 10 mA cm⁻² at an overpotential as low as 19 mV (Fig. 3a). Notably, the current density of Ru-WO₃₋ₓ/CP is enhanced by a factor of 24.0 as compared to the commercial Ru/C (5.0 wt.%)/CP at the

potential of −0.150 V vs. RHE. Meanwhile, the Tafel slope of Ru-WO₃₋ₓ/CP also significantly reduces to 41 mV dec⁻¹ (Fig. 3b), manifesting a change of the rate-determining step (RDS) of HER from water dissociation for Ru/C to hydrogen recombination for Ru-WO₃₋ₓ[46,47]. It is noteworthy that the Ru-WO₃₋ₓ/CP is among the best HER electrocatalysts reported in the neutral medium (Fig. 3c and Supplementary Table 8). Moreover, we also calculated the LSV curves normalized by the electrochemically active surface area and Ru-WO₃₋ₓ/CP still shows much better HER activity than Ru/C (5.0 wt.%)/CP (Supplementary Figs. 7 and 8). The HER activity and stability under higher current densities were also evaluated. As shown in Supplementary Fig. 9, Ru-WO₃₋ₓ/CP displays a low overpotential of 225 mV to achieve a current density of 1 A cm⁻², and the potential of Ru-WO₃₋ₓ/CP remains stable to attain 1 A cm⁻² in the chronopotentiometry test. Besides activity, the Ru-WO₃₋ₓ/CP also displays excellent durability in catalyzing HER (Fig. 3d). Both the structure and composition of Ru-WO₃₋ₓ/CP remain unchanged before and after the HER stability test as examined by SEM, HRTEM and ICP-OES (Supplementary Fig. 10 and Supplementary Table 7). Moreover, to explore whether oxygen vacancies have an effect on the HER activity, Ru-WO₃/CP was also synthesized for comparison. As shown in Supplementary Fig. 11, the HER activity of Ru-WO₃/CP is better than that of Ru/C (5.0 wt.%)/CP, but is still much worse than that of Ru-WO₃₋ₓ/CP. The corresponding Tafel plots indicate that the reaction kinetics of Ru-WO₃/CP is slower than that of Ru-WO₃₋ₓ/CP, but faster than that of Ru/C (5.0 wt.%)/CP. These results suggest that oxygen-deficient tungsten oxide is beneficial for activity improvement. Moreover, Ru-WO₃/CP keeps stable in the long-term stability test, and no obvious HER activity decay is observed. In addition, WO₃ or WO₃₋ₓ gradually dissolves in alkaline electrolyte[48,49]. Thus,

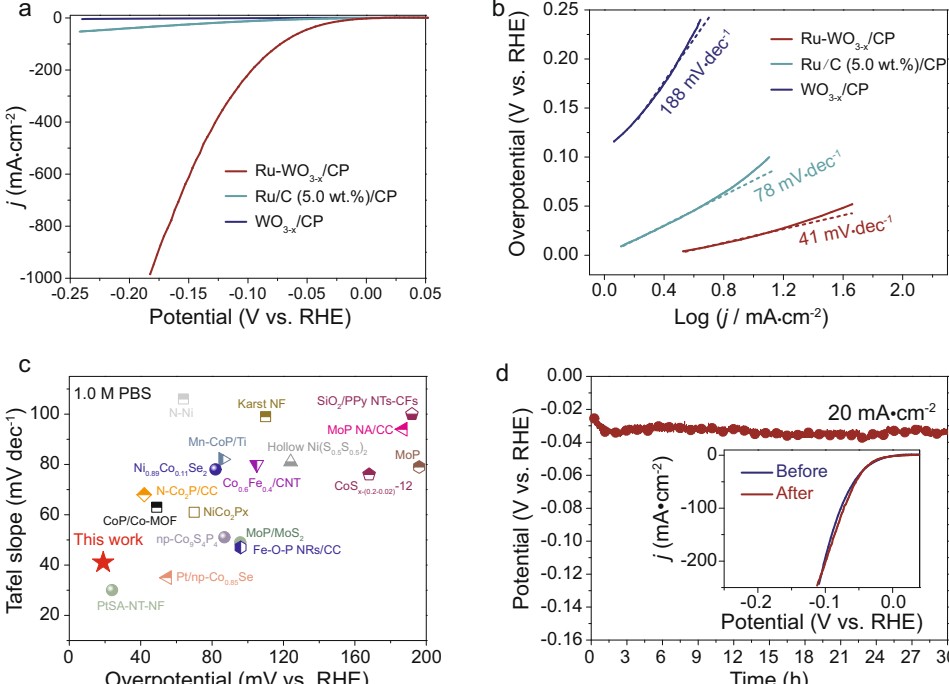

**Fig. 3 | HER performance in 1.0 M PBS. a** LSV curves in 1.0 M PBS. **b** The corresponding Tafel plots. **c** HER activity comparison of Ru-WO$_{3-x}$/CP with other reported state-of-the-art electrocatalysts in 1.0 M PBS. The corresponding references for these reported electrocatalysts are shown in Supplementary Table 8. **d** Chronopotentiometric curve recorded at a constant cathodic current density of 20 mA cm$^{-2}$. Inset compares the LSV curves before and after the stability test.

it is not appropriate to apply Ru-WO$_{3-x}$ in alkaline condition because of the stability issue.

To dig out the origin of the enhanced HER activity of Ru-WO$_{3-x}$/CP, we first examined the tungsten oxide support. According to previous reports, creating oxygen vacancies in tungsten oxide could increase its capacitance[40,41]. Since proton storage in tungsten oxide is positively correlated with its capacitive performance, gravimetric capacitance of Ru-WO$_{3-x}$, WO$_{3-x}$ and WO$_3$ were calculated from cyclic voltammetry (CV) measurements. Here, we separated the capacitive and diffusion-controlled contribution in the measured capacitance using the following equation: $i$ (V) = $k_1v + k_2v^{1/2}$, where $i$ (V), $k_1v$, and $k_2v^{1/2}$ are total current, capacitive current and diffusion-controlled current in CV, respectively[39]. The results at various scan rates show that the capacitance of WO$_{3-x}$ is enhanced by a factor of around 23.0 as compared to that of WO$_3$ (Supplementary Figs. 12–17). Loading Ru NPs onto WO$_{3-x}$ could further increase the capacitance to some extent. The proton insertion/extraction kinetics was assessed using the equation: $i = av^b$, where $i$ is the current response, $a$ is an adjustment coefficient, $v$ is the sweep rate and the power exponent b is a parameter to analyze the kinetics[50,51]. A $b$ value of 0.5 means diffusion-controlled kinetics and a $b$ value of 1.0 indicates an ideal capacitive or non-diffusion-controlled behavior. By plotting log ($i$) vs. log ($v$), the $b$ value of Ru-WO$_{3-x}$ is estimated to be 0.99 and 0.82 (Supplementary Fig. 13a, b), respectively, based on the oxidation and reduction redox peak, indicating the dominant pseudocapacitive behavior of proton extraction/insertion and that the proton extraction is more rapid than its insertion in Ru-WO$_{3-x}$.

To study the HER process, in situ Raman spectroscopy was performed. The in situ Raman spectra of Ru-WO$_{3-x}$/CP were collected in the potential range from −0.1 to −0.7 V (vs. Ag/AgCl), which includes both the non-Faradaic current region and the HER region. For the initial state at open circuit, the typical Raman peaks of WO$_{3-x}$ were observed at 778 cm$^{-1}$ (Fig. 4a), which is attributed to the W-O stretching vibration[45,52]. The Raman signal at 778 cm$^{-1}$ gradually decreased with increasing applied cathodic potential and this signal completely

disappeared at −0.5 V (vs. Ag/AgCl), due to proton insertion in WO$_{3-x}$[53,54]. Additionally, a new Raman peak at 878 cm$^{-1}$ appeared after the applied potential reached −0.6 V (vs. Ag/AgCl) and the intensity of this peak further increased with increase in applied cathodic potential. Based on previous reports, the Raman peak at 878 cm$^{-1}$ may result from the Ru-H stretching vibration[37,38]. To figure out the attribution of the Raman peak at 878 cm$^{-1}$, a deuterium isotopic substitution experiment was performed. Once H$_2$O was changed to D$_2$O, the Raman peak at 878 cm$^{-1}$ shifted to a lower wavenumber at 611 cm$^{-1}$ (Fig. 4b). The downward shift ratio ($\gamma$) in the isotopic substitution experiment can be estimated by: $\gamma = v(\text{Ru-D})/v(\text{Ru-H})$ (see details in Supplementary Note 2). The estimated downward shift ratio ($\gamma$) of the Ru-H peak is ∼71.0%, very close to the theoretical 70.0%. Furthermore, DFT calculation was performed to determine the vibrational frequency of H* on metallic Ru. Here, we considered two Ru models. The Raman frequency of the Ru-H stretching vibration was calculated to be 875 and 880 cm$^{-1}$ for the ridge and top sites on the Ru cluster (Fig. 4c). These results corroborate the attribution of the Raman peak at 878 cm$^{-1}$ to the Ru-H stretching vibration. When the potential was swept back from −0.6 V to −0.2 V (vs. Ag/AgCl) in 1.0 M PBS, the characteristic Raman peak of W-O stretching vibration re-appeared at −0.5 V (vs. Ag/AgCl) and the peak intensity increased with further decrease in the cathodic potential, suggesting that the proton extraction started at −0.5 V (vs. Ag/AgCl) (Fig. 4d). Notably, when the applied cathodic potential was increased, the Raman peak intensity of Ru-H first remained steady until −0.5 V (vs. Ag/AgCl) and then gradually increased, suggesting increase of hydrogen coverage on Ru, which may result from proton transfer from WO$_{3-x}$ to Ru NPs. Moreover, the deuterium isotopic substitution experiment should also be performed on WO$_{3-x}$/CP. As shown in Supplementary Fig. 18, there is no obvious difference between the Raman results in deuterium isotopic substitution and non-deuterium isotopic substitution experiments. Only W-O Raman peaks can be seen in the Raman spectra.

Furthermore, the hydrogen coverage ($\theta_H$) on Ru-WO$_{3-x}$/CP in HER was studied by microkinetic analysis (Supplementary Note 1,

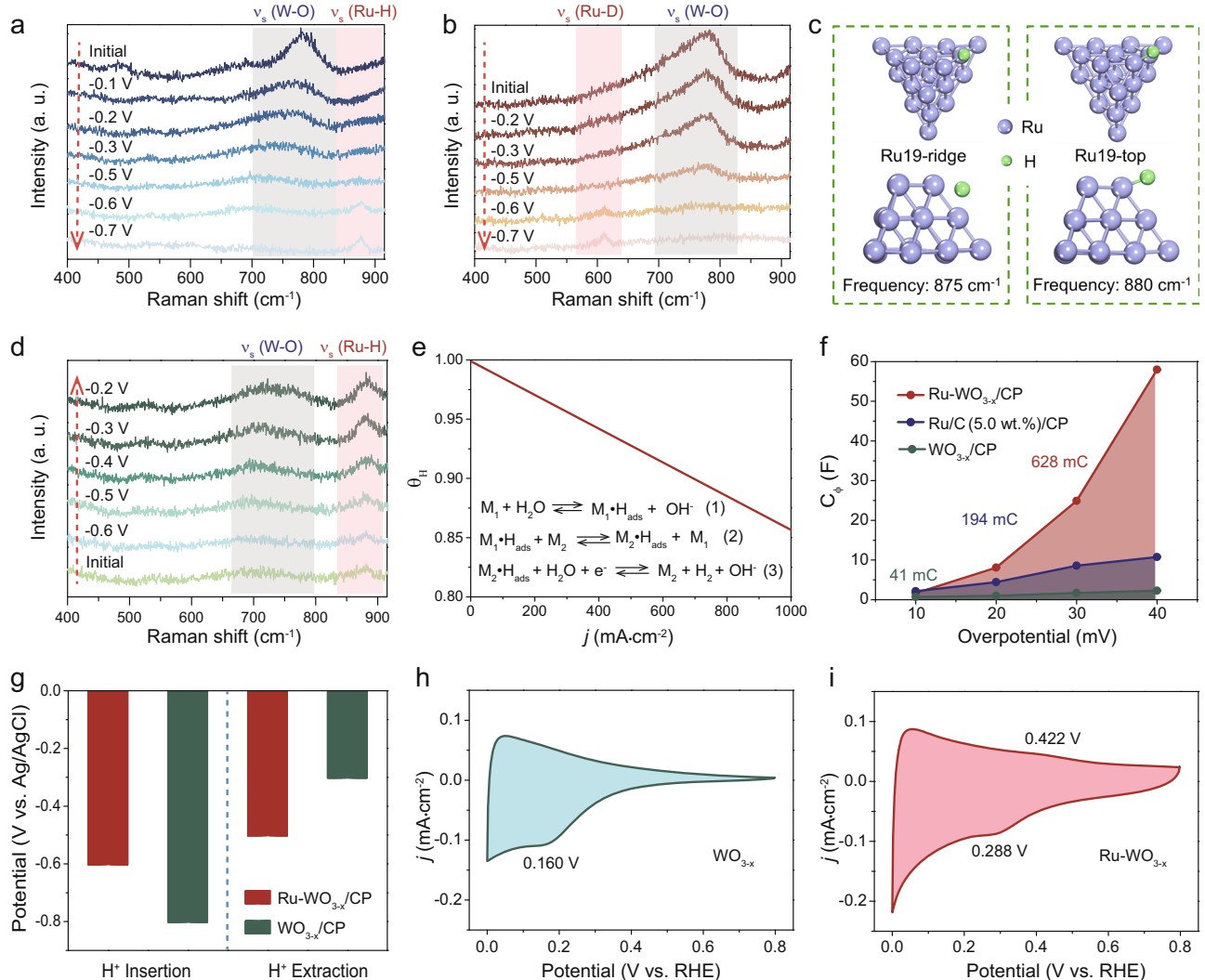

**Fig. 4 | Insights of hydrogen spillover. a** In situ Raman spectra of Ru-WO$_{3-x}$/CP recorded in 1.0 M PBS from −0.1 to −0.7 V vs. Ag/AgCl. **b** In situ Raman spectra of Ru-WO$_{3-x}$/CP recorded from −0.2 to −0.7 V vs. Ag/AgCl in 1.0 M PBS (in D$_2$O). **c** Side and top view illustrations of DFT models used for Raman frequency calculation of Ru-H vibration. **d** In situ Raman spectra of Ru-WO$_{3-x}$/CP recorded in 1.0 M PBS from −0.6 to −0.2 V vs. Ag/AgCl. **e** Hydrogen coverage as a function of current density obtained from microkinetic analysis. **f** Fitted data of C$_\varphi$ at different overpotentials for various electrocatalysts during HER in 1.0 M PBS. **g** The onset potential of H$^+$ insertion and extraction for Ru-WO$_{3-x}$/CP and WO$_{3-x}$/CP. **h** CV curve of WO$_{3-x}$ recorded in 1.0 M PBS. Scan rate: 5 mV/s. **i** CV curve of Ru-WO$_{3-x}$ recorded in 1.0 M PBS. Scan rate: 5 mV/s.

Supplementary Tables 1–6, and Supplementary Figs. 1 and 2). Compared with $\theta_H$ on Ru/C (5.0 wt.%)/CP (Fig. 1b), $\theta_H$ on Ru-WO$_{3-x}$/CP significantly increases (Fig. 4e), matching well with the in situ Raman spectra (Figs. 1c and 4a). Operando electrochemical impedance spectroscopy (EIS) measurement was further conducted to probe the hydrogen adsorption/desorption process in HER[27,53]. The obtained EIS data was simulated using an equivalent circuit model as shown in Supplementary Fig. 19 and Supplementary Table 9. The first parallel circuit, with one resistor ($R_c$) and one constant phase element (CPE$_1$), corresponds to the inner layer of electrode material, where $R_c$ is the charge transfer resistance and CPE$_1$ corresponds to the double-layer capacitance. The second parallel circuit simulates the electrolyte-catalyst interfacial charge transfer[27,55,56], which is able to reflect the hydrogen intermediate adsorption behavior on the catalytically active sites ($R_i$ is the charge transfer resistance and C$\varphi$ represents the hydrogen adsorption pseudo-capacitance). C$\varphi$ as a function of overpotential was integrated to calculate the hydrogen adsorption charge ($Q_{H^*}$). Compared with $Q_{H^*}$ for Ru/C (5.0 wt.%)/CP, $Q_{H^*}$ (originated from Ru) ($Q_{H^*(Ru-WO3-x/CP)} - Q_{H^*(WO3-x/CP)}$) for Ru-WO$_{3-x}$/CP increases a lot (Fig. 4f), in good agreement with

the greatly increased $\theta_H$ on Ru-WO$_{3-x}$/CP deduced from microkinetic analysis.

To explore factors that influence proton insertion/extraction in tungsten oxide, in situ Raman measurements were also performed on WO$_{3-x}$/CP (Supplementary Fig. 20). By comparing the onset potential of proton insertion/extraction, it is found that introduction of Ru onto WO$_{3-x}$ results in a more positive onset potential for proton insertion while a more negative onset potential for proton extraction (Fig. 4g). This indicates that Ru NPs can not only promote proton insertion but also accelerate proton extraction from WO$_{3-x}$. Furthermore, CV was performed in 1.0 M PBS to investigate the proton insertion/extraction behavior (Fig. 4h, i). In the CV curves, obvious proton insertion peaks are observed on WO$_{3-x}$ at 0.160 V (vs. RHE) and Ru-WO$_{3-x}$ at 0.288 V (vs. RHE)[39,40]. Furthermore, a distinct proton extraction peak at 0.422 V (vs. RHE) can be observed on Ru-WO$_{3-x}$, but not on WO$_{3-x}$, strongly illustrating that Ru NPs can promote proton extraction from WO$_{3-x}$.

## Theoretical insights of reversible hydrogen spillover in HER
To gain some theoretical insights on whether hydrogen spillover can take place from WO$_{3-x}$ to Ru NPs, DFT calculations were carried out to

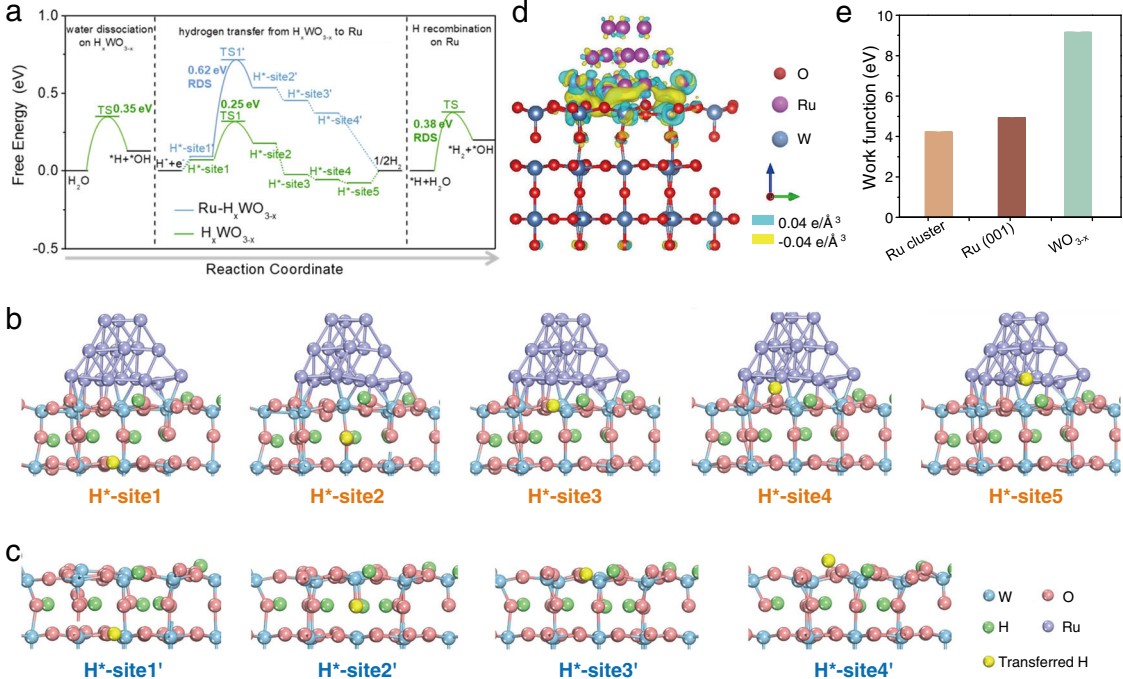

**Fig. 5 | DFT calculations of hydrogen transfer energy barrier. a** Calculated free energy diagram for HER on Ru-$H_xWO_{3-x}$ and $H_xWO_{3-x}$. **b** The optimized H* adsorption structure on various sites of Ru-$H_xWO_{3-x}$. **c** The optimized H* adsorption structure on various sites of $H_xWO_{3-x}$. **d** Electron density difference plot across the Ru-$WO_{3-x}$ interface. Electron accumulation and depletion are indicated in yellow and blue, respectively. **e** Work function calculations for various Ru and $WO_{3-x}$.

determine the hydrogen transfer energy barrier. The Gibbs free energy change at each step of hydrogen transfer from the interior of $WO_{3-x}$ to the surface of Ru NPs was computed. As hydrogen spillover occurs on proton-inserted $WO_{3-x}$ ($H_xWO_{3-x}$), a moderate amount of H was added into the as-built $WO_{3-x}$ model to simulate the actual situation. As shown in Fig. 5a–c, hydrogen adsorption is extremely weak on the external surface of $H_xWO_{3-x}$ (site 2, 3 and 4), while it becomes much stronger on the inner surface (site 1), suggesting that interior of $H_xWO_{3-x}$ is more favorable for hydrogen adsorption. The difference of hydrogen adsorption between internal and external surface results in a large difference in Gibbs free energy of adsorbed hydrogen ($\Delta G_{H^*}$) between site 1 and site 2, with a thermodynamic barrier of 0.44 eV. Meanwhile, the kinetic barrier of hydrogen transfer from site 1 to site 2 was calculated to be 0.62 eV, indicating difficulty of hydrogen transfer from site 1 to site 2. On the contrary, $\Delta G_{H^*}$ values of site 2, 3 and 4 on Ru-$H_xWO_{3-x}$ become more negative, suggesting much improved hydrogen adsorption on the external surface of Ru-$H_xWO_{3-x}$. Benefitted from the enhanced hydrogen adsorption, the thermodynamic barrier of hydrogen transfer from site 1 to site 2 reduces to 0.11 eV. Furthermore, the kinetic hydrogen transfer barrier (from site 1 to site 2 on Ru-$H_xWO_{3-x}$) was calculated to be 0.25 eV, manifesting that the hydrogen transfer process is greatly promoted on Ru-$H_xWO_{3-x}$. Additionally, it is found that the hydrogen adsorption on Ru surface (site 5) is stronger than that on $H_xWO_{3-x}$, as indicated by the most negative $\Delta G_{H^*}$ of −0.078 eV on site 5. Therefore, the adsorbed hydrogen can be spontaneously transferred from external surface of $H_xWO_{3-x}$ to Ru (from site 4 to site 5). To unravel the facilitated hydrogen transfer process on Ru-$WO_{3-x}$, the charge density difference was calculated to explore the charge distribution at the interface. As shown in Fig. 5d, electron accumulation is observed below the surface layer of $WO_{3-x}$. High density electrons are favorable to trap hydrogen atoms via interacting with unsaturated electrons in the H 1s orbital. Therefore, hydrogen adsorption is significantly enhanced on the external surface of $WO_{3-x}$. Moreover, electron accumulation below the surface layer of

$WO_{3-x}$ also attracts protons in the interior to the surface by electrostatic interaction. As a result, hydrogen spillover from $WO_{3-x}$ to Ru is thermodynamically and kinetically facilitated. Next, to investigate the cause of charge transfer between Ru NPs and $WO_{3-x}$, the work functions (φ) of Ru and $WO_{3-x}$ were calculated. The work function of Ru NPs was determined to be 4.91 eV, much smaller than that of $WO_{3-x}$ (9.14 eV), revealing electron transfer from Ru to $WO_{3-x}$ (Fig. 5e and Supplementary Figs. 21 and 22). Taking into account the size distribution of Ru NPs, the work function of bulk Ru was also computed, which is slightly larger than that of Ru NPs but still much smaller than that of $WO_{3-x}$. Combining with the above analyses, a reasonable explanation for hydrogen spillover from $WO_{3-x}$ to Ru is given as follows: the difference in work function between Ru NPs and $WO_{3-x}$ leads to electron accumulation at the subsurface of $WO_{3-x}$, which enhances hydrogen adsorption and also drives moving internal protons to the external surface. In addition, DFT calculations were also performed to reveal the free energy change in H transfer and the electronic structure of Ru-$WO_3$/CP (Supplementary Fig. 23). The free energy change in H transfer of Ru-$WO_3$/CP is comparable to that of Ru-$WO_{3-x}$/CP, and the kinetic energy barrier of Ru-$WO_3$/CP from site 1 to site 2 is determined to be 0.31 eV, only slightly higher than that of Ru-$WO_{3-x}$/CP, indicating that H transfer is also favorable on Ru-$WO_3$/CP. The charge difference of Ru-$WO_3$/CP also reveals that electrons transfer from Ru to $WO_3$, which is the same as that of Ru-$WO_{3-x}$/CP.

To explore the RDS of HER on Ru-$H_xWO_{3-x}$, we conducted DFT calculations to investigate the energy barrier of reaction steps. It is well-known that water dissociation is the RDS in alkaline and neutral media water oxidation, which can be deduced from the very large Tafel slopes. The Heyrovsky reaction and Tafel reaction are closely correlated with the Gibbs free energy of adsorbed hydrogen ($\Delta G_{H^*}$), which is the typical HER descriptor. We calculated the energy barrier of water dissociation and the $\Delta G_{H^*}$ on various sites to explore the RDS of HER taking place on Ru-$H_xWO_{3-x}$. As revealed in Supplementary Figs. 24–26, the barrier for water dissociation on Ru is about 0.46 eV

and that on $WO_{3-x}$ is around 0.35 eV. The water dissociation on $WO_{3-x}$ is more favorable than that on Ru. As the further increase of over-potential, $WO_{3-x}$ will dissociate the water to generate protons, which can also spillover to Ru. The OH on the surface of $WO_{3-x}$ will undergo desorption and then be quickly captured by the buffer electrolyte. The OH concentration on the catalyst's surface is low because the buffer electrolyte can react quickly with the desorbed OH to produce $H_2O$. As shown in Supplementary Figs. 27 and 28, the $\Delta G_{H^*}$ on the corner and edge sites of Ru clusters of Ru-$H_xWO_{3-x}$ are close to 0 eV, which is the thermodynamically neutral state. To further explore the reaction process, we calculated the energy barriers of the Tafel step and Heyrovsky step on Ru, respectively. As shown in Supplementary Figs. 29–31, the Heyrovsky step shows a lower energy barrier than the Tafel step, suggesting that the hydrogen atoms on Ru transferred from $WO_{3-x}$ tend to follow the Heyrovsky mechanism to generate di-hydrogen. As the energy barrier of hydrogen transfer from $WO_{3-x}$ to Ru is only 0.25 eV, the whole process to generate di-hydrogen including hydrogen atoms spillover from $WO_{3-x}$ to Ru followed by the Heyrovsky step to generate di-hydrogen has an energy barrier of 0.38 eV, indicating Heyrovsky step is the rate limiting step in this process (Fig. 5a). However, the di-hydrogen produced by water dissociation over Ru itself has a higher energy barrier of 0.46 eV, suggesting that this process is more difficult than the spillover mechanism to generate molecular hydrogen. Therefore, the H atoms involved in the reaction tend to come from H transfer from $H_xWO_{3-x}$ to Ru rather than water dissociation on Ru of Ru-$WO_{3-x}$/CP.

Based on the above investigations, a possible mechanism is proposed to account for the greatly enhanced HER activity of Ru-$WO_{3-x}$ in neutral medium (Fig. 6). Under applied cathodic potential, protons in the electrolyte were inserted into the oxygen-deficient $WO_{3-x}$ and then coupled with electrons to form $H_xWO_{3-x}$. The plenty of oxygen vacancies in $WO_{3-x}$ significantly increased the proton storage capacity and at the same time improved the charge transfer. As a result, the oxygen-deficient $WO_{3-x}$ served as a proton reservoir to supply protons onto Ru surface, which recombined to evolve molecular hydrogen. As the further increase of overpotential, $WO_{3-x}$ will dissociate the water to generate protons, which also spilled over to Ru. The hydrogen spillover from $WO_{3-x}$ to Ru changed the RDS of HER on Ru in neutral medium from water dissociation to hydrogen recombination, which greatly improved the HER kinetics (denoted as pathway 1). In addition, hydrogen recombination on $H_xWO_{3-x}$ (denoted as pathway 2) and hydrogen formation on Ru NPs (denoted as pathway 3), where hydrogen is all provided by water dissociation on Ru, are both unfavorable.

## Discussion

In summary, we report the design and performance of Ru-$WO_{3-x}$ to facilitate different parts of the multistep HER process in neutral environment: the oxygen-deficient $WO_{3-x}$ possesses a large capacity for storing protons, which can be transferred to the surface of Ru NPs under cathodic potential. This hydrogen spillover from $WO_{3-x}$ to Ru changes the RDS of HER on Ru in neutral medium from water dissociation to hydrogen recombination, which greatly improves the HER kinetics.

## Methods
### Materials
Ruthenium (III) chloride anhydrous ($RuCl_3$) and ammonium meta-tungstate hydrate (($NH_4$)$_6H_2W_{12}O_{40}$·$xH_2O$) with 99.5% metals basis were purchased from Shanghai Macklin Biochemical Co., Ltd. Potassium phosphate monobasic ($KH_2PO_4$, AR, 99.5%) and potassium hydrogen phosphate ($K_2HPO_4$, ≥98%, ACS) were obtained from Shanghai Aladdin Bio-Chem Technology Co., Ltd. The TGP-H-060 TORAY carbon paper (CP) was purchased form Suzhou Sinero Technology Co., Ltd. The ethanol ($CH_3CH_2OH$, AR, ≥99.5%) and acetone ($C_3H_6O$, AR, ≥95%) were purchased from Sinopharm Chemical Reagent Co., Ltd. Nickel(II) acetate tetrahydrate (Ni(O-COCH$_3$)$_2$·4H$_2$O, AR, 98%), chloroplatinic acid hexahydrate ($H_2PtCl_6$·6H$_2$O, ACS reagent, ≥37.50% Pt basis), sodium borohydride ($NaBH_4$, granular, 99.99% trace metals basis), sodium hydroxide (NaOH, ACS reagent, ≥97.0%, pellets), and molybdenum(VI) oxide ($MoO_3$, nanopowder, 100 nm (TEM), 99.5% trace metals basis) were purchased from Sigma-Aldrich. All chemicals were used as received with purification. Ultrapure water (deionized) was used in all the experiments.

### Preparation of $WO_3$/CP
A piece of CP (4 cm × 3 cm × 0.19 mm) was cleaned by ultrasonication in acetone for 15 min, and then dried in air. ($NH_4$)$_6H_2W_{12}O_{40}$·$xH_2O$ (4.0 mmol, 11.825 g) was dissolved in 60.0 ml ultrapure water, followed by magnetic stirring for 20 min to obtain a homogeneous solution. The above solution was transferred into a 100 ml autoclave and CP was placed vertically in the solution. Then, the autoclave was heated at 180 °C for 16 h. The obtained $WO_3$/CP was washed in deionized water and dried at 70 °C overnight.

### Preparation of $WO_{3-x}$/CP
$WO_3$/CP (1 cm × 1.5 cm) was annealed at 400 °C for 2 h at a heating rate of 5 °C/min in $H_2$/Ar (10/90) atmosphere to obtain $WO_{3-x}$/CP.

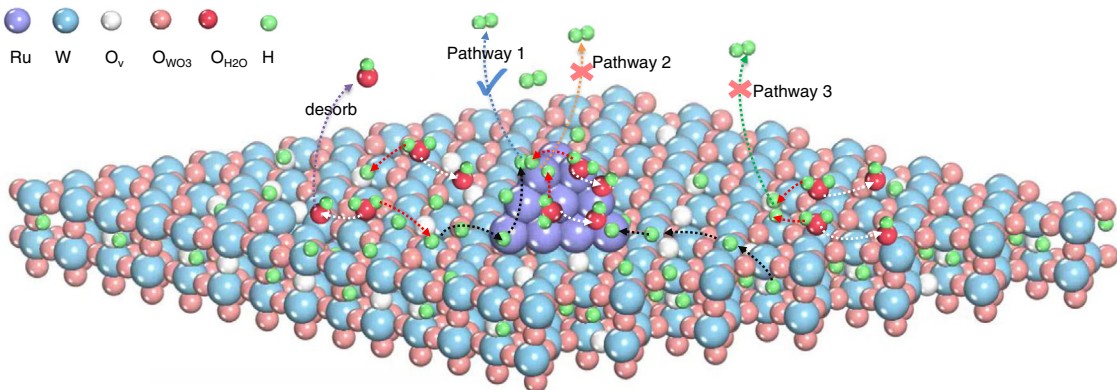

**Fig. 6 | Illustration of hydrogen spillover.** Schematic diagram showing how hydrogen spillover from $WO_{3-x}$ to Ru enhances HER in neutral environment. The white and red arrows indicate the formation process of OH and H from the dissociation of water, respectively. Black arrows indicate the transfer process of H. The purple arrow indicates the desorption of OH adsorbed on the surface. The blue, orange and green arrows indicate the different ways of formation of molecular hydrogen.

## Preparation of Ru-WO₃₋ₓ/CP

RuCl₃ (40.0 mg) was dissolved in 20.0 ml deionized water by magnetic stirring and ultrasonication to obtain 2.0 mg/ml RuCl₃ solution. WO₃/CP (1 cm × 1.5 cm) was immersed in 10 ml 2.0 mg/ml RuCl₃ solution for 2 min. Subsequently, the WO₃/CP was taken out and dried in an infrared desiccator for 8 min to obtain RuCl₃-WO₃/CP. Finally, the RuCl₃-WO₃/CP was annealed at 400 °C for 2 h at a heating rate of 5 °C/min in H₂/Ar (10/90) atmosphere to obtain Ru-WO₃₋ₓ/CP.

## Preparation of Ru-WO₃/CP

NaBH₄ (100.0 mg) was dissolved in 20.0 ml 1.0 M NaOH solution under magnetic stirring to obtain 5.0 mg/ml NaBH₄ solution. WO₃/CP (1 cm × 1.5 cm) was immersed in 10 ml 2.0 mg/ml RuCl₃ solution for 2 min. Subsequently, the WO₃/CP was taken out and dried in an infrared desiccator for 8 min to obtain RuCl₃-WO₃/CP. Finally, RuCl₃-WO₃/CP was immersed in 5.0 mg/ml NaBH₄ solution for 10 min to obtain Ru-WO₃/CP.

## Materials characterizations

Power XRD patterns were recorded on a Rigaku Ultima IV (Cu Kα radiation, $\lambda = 1.54$ Å) diffractometer at the operating voltage of 40 kV and current of 20 mA. The HRTEM measurement was conducted on a JEOL JEM-2100F with a 200 kV acceleration voltage. XPS spectra were collected on a Thermo Scientific ESCALAB 250Xi with Al Kα radiation (1486.6 eV). SEM images were taken on a Hitachi SU8010 microscope. Raman spectroscopy analysis was performed on a JY, HR 800 Raman spectrometer with a 514 nm laser. EPR experiment was conducted on a Bruker A300-10/12. Oxygen temperature programmed desorption (O₂-TPD) analysis was performed on a BELCAT II fully automatic chemisorber instrument (MicrotracBEL). The procedures were as follows: (1) each sample was pretreated under a He flow (50 ml min⁻¹) at 300 °C for 30 min; (2) the sample was purged with 5% O₂/He for 1 h at 50 °C for O₂ adsorption; (3) the sample was heated to 700 °C at a heating rate of 10 °C min⁻¹ under a pure He gas flow. The signal of O₂ desorption was measured by a thermal conductivity detector.

## Electrochemical measurements

All of the electrochemical measurements were conducted on a CHI 760E electrochemical workstation (CH Instruments Ins.) in a three-electrode system at room temperature. The geometric area of the CP is 1 cm × 1 cm. A graphite rod electrode and a saturated calomel electrode (SCE) were used as the counter electrode and the reference electrode, respectively. The potential of SCE vs. reversible hydrogen electrode (RHE) was determined by performing CV scans (scan rate: 1 mV/s) in a hydrogen-saturated electrolyte with a Pt plate as both the working and counter electrode, and the average value of the two potentials at the current of zero in the CV curve is regarded as the potential of SCE vs. RHE. Linear sweep voltammetry (LSV) was performed at a scan rate of 2 mV s⁻¹ after purging H₂ in the electrolyte for 20 min. All of the potentials in LSV are iR-corrected. The resistance for iR-compensation was tested at the open circuit potential. EIS was conducted in the frequency range from 10⁵ Hz to 10⁻³ Hz. All potentials are referenced to the RHE by the Nernst equation: $E_{(RHE)} = E_{(SCE)} + 0.0591 \times pH + 0.242$ V, unless otherwise stated. In total, 1.0 M phosphate buffered solution (PBS) was prepared by mixing 1.0 M K₂HPO₄ with 1.0 M KH₂PO₄ in a volume ratio of 2:1. In this work, only the potentials in the Raman spectra are relative to the Ag/AgCl electrode; all other potentials are relative to the RHE.

## CV measurements to determine the specific capacitance

As CP has double-layer capacitance, the powders of WO₃, WO₃₋ₓ and Ru-WO₃₋ₓ, which were peeled off from WO₃/CP, WO₃₋ₓ/CP and Ru-WO₃₋ₓ/CP, were used for the CV measurements. The ink for the working electrode was prepared by dispersing 5 mg of catalyst in a mixture of 480 μl of ethanol and 20 μl of 5 wt% Nafion solution, followed by sonication for 30 min to obtain a homogeneous dispersion. A 10 μl of the ink was cast on the glassy carbon electrode (5 mm of diameter) and then dried in air; the as-prepared electrode served as the working electrode. A SCE and a graphite rod electrode were used as the reference electrode and the counter electrode, respectively. All of the CV scans were performed after purging N₂ into the electrolyte for 20 min.

## In situ Raman spectroscopy measurements

In situ Raman spectra were recorded on a LabRAM HR Evolution (HORIBA Scientific) spectrometer. The electrochemical cell used for Raman measurement was homemade by Teflon and a quartz plate was employed as the window to cross the laser. A Pt wire and an Ag/AgCl electrode (1.0 M KCl as inner filling electrolyte) were applied as the counter electrode and the reference electrode, respectively. To apply a controlled potential on the catalyst during the Raman measurement, chronoamperometry was performed at various potentials in 1.0 M PBS. The illustration of operando Raman spectroscopy setup is shown in Supplementary Fig. 32.

## DFT calculation

Density functional theory (DFT) calculations using the plane-wave technique were conducted in the Vienna Ab Initio Simulation Package. The exchange-correlation functional was the Perdew-Burke-Emzerhof parametrization of the generalized gradient approximation. The electron-ion interactions were described by the projector augmented wave. Van der Waals interactions were corrected by the DFT-D3 method. A plane-wave basis was set with the cutoff energy of 400 eV. The Brillouin zone was built with a (2 × 2 × 1) Monkhoest-Pack k-point mesh for all models in the optimization of the supercell structure. The force residue for relaxation of all the atoms was set as 0.02 eV/Å. The lattice parameter used for hexagonal WO₃ was 7.51 Å × 7.51 Å × 7.71 Å (a × b × c). Gibbs free energy of hydrogen adsorption was calculated by:

$$\Delta G_H = E_{H/surf} - E_{surf} - 1/2 E_{H2} + \Delta E_{ZPE} - T\Delta S_H$$

where $E_{H/surf}$ is the total energy of surface with adsorbate, $E_{surf}$ is the energy of clean surface, $E_{H2}$ is the energy of a gas phase H₂ molecule, $\Delta E_{ZPE}$ represents the zero-point energy of the system and was taken as 0.05 eV, and $T\Delta S_H$ is the contribution from entropy and was simplified as 0.20 eV at 298 K. In this work, the final state of the calculated H₂ is in gas phase.

## Reporting summary

Further information on research design is available in the Nature Research Reporting Summary linked to this article.

# Data availability

The remaining data contained within the paper and Supplementary Files are available from the authors upon request.

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

## Acknowledgements

We are grateful for the financial support from the National Key R&D Program of China (2021YFB3801600), the National Natural Science Foundation of China (21872121, 21908189), the Fundamental Research Funds for the Central Universities (2017XZZX002–16), Ministry of Education of Singapore (Tier 1: RG4/20 and Tier 2: MOET2EP10120-0002), and Agency for Science, Technology and Research (AME IRG: A20E5c0080).

## Author contributions

Y.W. and B.L. conceived the study. J.C., C.C. and M.Q. designed the experiment and performed the initial tests. J.C., Q.M., B.Lin, B.Li and H.Y. conducted the theoretical calculations. J.C., C.C. and M.Q. wrote the manuscript. All authors discussed the results and commented on the manuscript.

## Competing interests

The authors declare no competing interests.
