## [Peer Review File · Nature Communications]

Reversible hydrogen spillover in Ru-WO₃-x enhances hydrogen evolution activity in neutral pH water splittingREVIEWER COMMENTS

Reviewer #1 (Remarks to the Author):

The authors propose the hydrogen spillover from $\text{WO}(3-x)$ to Ru NPs during the hydrogen evolution reaction based on results obtained using a variety of techniques (in situ Raman spectroscopy, scan potentials and impedance measurements, complemented by DFT calculations).

First, the interpretation of catalytic effects through the spillover phenomenon is not new and nowadays no more groundbreaking... The term spillover was introduced in the sixties to explain just the migration of hydrogen atoms from platinum to WO_3 used as support. Since the nineties there is a plenty of works ... it was a controversial topic because of the lack of a scientific basis for discussion. Recently, there was a revival. There is an excellent review (not cited by the authors) by R. Prins (Chem. Rev. 2012, 112, 2714–2738) where the facts and fiction are discussed. Therefore, I don't think, this work is suitable for publication in Nature Communications.

Second, although the exhaustive analysis performed by the authors, I do not coincide with the interpretations.

According to a reference mentioned by the authors (Angew. Chem., Int. Ed. 58, 16038-16042 (2019)) the tendency is the migration of hydrogen atoms from Pt to $\text{WO}(3-x)$. Why should go in the opposite direction as proposed in this work?

Where does the hydrogen in $\text{WO}(3-x)$ come from? the dissociation of water (according to the calculations of the authors) is about 0.87 eV!! the value of this barrier for the dissociation on Ru cluster is not given...

The authors estimate a capacitance enhancement of about 20 for the system $\text{WO}(3-x)$. This means a much larger real area for this system. Could be the enhancement of also about 20 observed for the HER caused by this increase in the area? a larger roughness factor?

Minors details: the authors give sometimes values of potential vs RHE and sometimes vs Ag/AgCl according to the electrode reference used in each experiment. It should be clearer for the reader if the values are informed always relative to the same reference electrode.

Reviewer #2 (Remarks to the Author):

Letter to the authors

The present manuscript by Chen, Liu, Wang and colleagues compares Ru catalysts for the HER reaction in neutral pH water splitting in the presence of hydrogen treated tungsten oxide. A significant increase in activity is observed in the presence of tungsten oxide. The authors ascribe the increase in activity by looking at hydrogen spillover. The authors provide a range of experimental activity measurements and characterization tools. In addition, the authors provide theoretical calculations to explore their experimental findings.

Even though, similar findings have been reported before using Pt instead of Ru (Angew. Chemie Int. Ed. 58, 16038–16042, 2019), I think that the work has the potential to provide essential mechanistic insights to warrant the publication in Nature Communications. However, in the current state the manuscript lacks a correct description of the involved spillover chemistry and the attempted theoretical explanation is not considering the most crucial step – the water dissociation reaction. If this the concept of two active structures (water activation on a metal oxide and combination/desorption on a metal) is convincingly shown, an important mechanistic path for HER is shown.

I suggest the publication of the manuscript, after the authors provided convincing theoretical insights into the different pathways of water dissociation for tungsten oxide and Ru. In addition, the should

revise their wording and description of the involved chemistry by considering the existing literature on hydrogen spillover (which is currently not appearing in the manuscript).

This is detailed in the following comments.

Major comments

1. Reaction mechanism: The authors report that the rate limiting process for Ru/C catalysts is water dissociation. They observe a change in the rate limiting step (rls) when adding reduced "WO_{3-x}" to the catalyst. Their kinetic modelling and in situ Raman spectroscopy suggests that H combination and desorption is the rls for this catalyst under their testing conditions. Now, the authors describe "WO_{3-x}" as a proton storage medium, which increases the H coverage of Ru. This discussion misses the relevant point: Why is the water dissociation reaction not the rls in the presence of "WO_{3-x}"? The authors do not provide a satisfying discussion and theoretical evaluation of this question. To me, it appears that the reduced "WO_{3-x}" is now acting as a catalyst for water dissociation (which is faster than Ru) and Ru mainly acts as H combination catalyst.

Therefore, the theoretical calculation do not discuss the relevant process and provide no explanation why Ru/WO_{3-x} performs better than Ru/C. The calculated spillover process of noble metals to or from tungsten oxide have been calculated and discussed before (see below) for heterogeneous catalysis and the presented theoretical results (H can hop from WO₃ to Ru) do not appear novel. The authors should compare the dissociation of water and a Ru (NP) surface vs. the dissociation on "WO_{3-x}" with theoretical calculation. This would provide significant new insights in the interplay of different active centers.

2. Discussion of spillover: Noble metal/WO₃ catalysts is the archetypical system for hydrogen spillover (first reported 1964). In the classical definition hydrogen spillover describes the dissociative adsorption of di-hydrogen on a (metal)-surface and the subsequent spillover to a surface (often a metal oxide) which is kinetically not able to adsorb di-hydrogen on its own. The reversible process (hydrogen combines on a metal surface and desorbs) is generally called reversible spillover. The authors describe this process here. Consequently, the discussion should be adopted to this.

In this respect, the authors should consider literature which currently appears to be not considered. The field of heterogeneous (thermal) catalysis provides an extensive report on spillover, amongst others:

- Khoobiar, S. Particle to Particle Migration of Hydrogen Atoms on Platinum—Alumina Catalysts from Particle to Neighboring Particles. *J. Phys. Chem.* 68, 411–412 (1964).
- Miu, E. V. & McKone, J. R. Comparisons of WO₃ reduction to H: XWO₃ under thermochemical and electrochemical control. *J. Mater. Chem. A* 7, 23756–23761 (2019).
- Prins, R. Hydrogen Spillover. *Facts and Fiction. Chem. Rev.* 112, 2714–2738 (2012).
- Levy, R. & Boudart, M. The kinetics and mechanism of spillover. *J. Catal.* 32, 304–314 (1974).
- Boudart, M. 'On the nature of spilt-over hydrogen'. *J. Mol. Catal. A Chem.* 138, 319–321 (1999).

Also, I recommend the change of the title. This is by no means "unconventional", a more suitable title appears to be "Reversible hydrogen spillover from WO_{3-x} to Ru in Ru-WO_{3-x} enhances hydrogen evolution activity in neutral pH water splitting".

3. Discussion of the tungsten oxide structure: Hydrogen treatments of WO₃ as know to produce H_xWO₃ bronzes (see literature). This is especially true for WO₃ in contact with a noble metal. The authors do a hydrogen pretreatment and consequently describe the materials "oxygen-deficient WO_{3-x}", this appears to be an assumption that is not justified by the authors experiments. EPR (line 109-111) and XPS (line 116-121) is indicative that W(VI) is transformed to W(V) but do not provide the evidence an evidence that indeed vacancies are formed. Hydrogen treatment will also form the H_xWO₃ bronze which will lead to a similar observed reduction of W. Therefore, the authors should

amend their discussion to this fact and should consider to use H_xWO_3-y instead of WO_3-x throughout their manuscript.

Minor comments

- Theoretical calculations: What is the final state of the calculated H_2 ? Gas phase or in solution? The desorption/adsorption in liquid media is highly complex and is therefore hard to model (e.g. *Angew. Chemie Int. Ed.* 58, 3527–3532, 2019). The authors should be specific which final state of H_2 was assumed/calculated.
- line 70: “we need to find ways to increase θ_H .” This seems not correct; increasing the coverage is the consequence of enhancing the water dissociation reaction, but should not be the goal. I suggest to revise the sentence.
- line 120: “In O1s XPS spectrum, two deconvoluted peaks centered at 531.70 eV and 530.30 eV can be assigned to oxygen vacancy and lattice oxygen, respectively.” Even though, such assignment is reported, such assignment is incorrect. The XPS signal of the O1s originates from the electrons of an oxygen atom. An oxygen vacancy has no oxygen atom from which the signal can come from. Therefore, this second peak can have other origins: The presence of OH groups or a lattice oxygen bound to a W(V) atom (close to a vacancy). Both would cause a shift in binding energy compared to lattice oxygen in WO_3 . The authors should change the discussion and wording.
- Based on the (S)TEM measurements, the authors should make a statement on the size of the Ru nanoparticles.
- The introduction (line 33) should provide more details on advantages/disadvantages of alkali vs. acidic conditions in water splitting.
- Line 26: “to meet the energy dilemma” It is not clear what the “energy dilemma” is. I suggest rephrasing.
- Figure 4b: Raman vibration frequency should be labeled with Ru-D (or Ru-2H) not Ru-H.
- The materials and method section is reporting the synthesis protocols for Pt, Ni- WO_3 and Ru- MoO_3 catalysts, which were not discussed in the manuscript. The performance of these materials is clearly different but may be presented in another manuscript. The authors should remove this section or discuss the data in the manuscript.
- Line 309: This is not the Discussion section; this is the conclusion/summary. I suggest to remove the heading or change to conclusion. The authors discuss along with the presentation of the results. The section (line 57) should be labelled “Results and discussion”)

Reviewer #3 (Remarks to the Author):

In this work, Chen et al. integrated Ru NPs on oxygen-deficient WO_3-x for HER in neutral electrolyte, and proposed that protons inserted into WO_3-x can be transferred to Ru NPs during HER. The obtained Ru- WO_3-x enhances HER activity by a factor of 24.0 as compared to the commercial Ru/C. They also indicated the hydrogen spillover phenomenon through combined in situ Raman spectroscopy investigations, electrochemical measurements and DFT calculations. Overall, this work explained the hydrogen spillover mechanism in detail, however, there are a list of issues that should firstly be addressed prior to consideration for publication in *Nat. Commun.*

1. The latest research progress of H spillover and application prospect of neutral HER should be discussed and added in the Introduction.
2. The XRD peaks of Ru- WO_3-x in Fig. 2b show obvious shift to higher angles, while the authors did not explain it.
3. The WO_3-x reference sample should be well characterized, especially for oxygen vacancies.
4. In Fig.4d, the Raman peak intensity of Ru-H firstly remained steady until -0.5 V (vs. Ag/AgCl) and then gradually increased when more positive potentials were applied. The authors explained that it was due to the increase of hydrogen coverage on Ru. However, Fig. 4f shows that hydrogen coverage increases with the enhancing overpotentials. This is contradictory, please explain.

5. The deuterium isotopic substitution experiment should also be performed on WO₃-x.
6. The illustration of operando Raman spectroscopy setup should be provided in SI.
7. The Tafel slopes were analyzed at different potential regions for three samples. Noted that Tafel slopes show differences in low- and high-potential regions (see J. Am. Chem. Soc. 2022, 144, 6028–6039).
8. Ru-WO₃-x/CP reaches a j of 10 mA cm⁻² at an overpotential of 19 mV. Although it is superb, ampere-level current densities with relatively low cell voltages should be more meaningful. The paper (J. Am. Chem. Soc. 2022, 144, 6028–6039) proposed that hydrogen spillover-bridged Volmer/Tafel processes can enable 3.5 A cm⁻² under an overpotential of 251 ± 3 mV in 1.0 M KOH electrolyte. The HER activity and stability test of Ru-WO₃-x under higher current densities should be conducted to study the effect of hydrogen spillover in this system.
9. Considering the application of alkaline electrolyser, how about the performance of Ru-WO₃-x in alkaline electrolyte?
10. Could the limiting steps of the work, as suggested through simulations, be supported through the determination of kinetic isotope effects using D₂O?
11. The author said that "The second parallel circuit simulates the electrolyte-catalyst interfacial charge transfer, which is able to reflect the hydrogen intermediate adsorption behavior on the catalytically active sites." Please add some representative references after this conclusion.
12. The authors highlight that the H atoms involved in the reaction tend to come from H transfer from H_xWO₃-x to Ru rather than water dissociation on Ru-WO₃-x/CP. The supply pathway of H on H_xWO₃-x should be given and explained in DFT calculation.
13. The authors indicate that protons were firstly inserted into the oxygen-deficient WO₃-x. Does these protons come from water or PBS electrolyte? In addition, in the microkinetic model, the adsorption state of OH⁻ should be considered.
14. Please carefully check this manuscript to revise some mistakes, such as "Fig. out in line 173", "29, 3.0 in line 220", "alkaline and neutral media water oxidation in line 274".

Reviewer #1 (Remarks to the Author):

The authors propose the hydrogen spillover from $\text{WO}_{(3-x)}$ to Ru NPs during the hydrogen evolution reaction based on results obtained using a variety of techniques (in situ Raman spectroscopy, scan potentials and impedance measurements, complemented by DFT calculations).

First, the interpretation of catalytic effects through the spillover phenomenon is not new and nowadays no more groundbreaking...

Response: We thank the reviewer for the comment. We agree with the reviewer that the interpretation of catalytic effects through the spillover phenomenon has been extensively studied in thermal catalysis, but the spillover effect in electrochemical process is so far still in its infancy. Although there have been some recent reports discussing the hydrogen spillover effect in HER (*Nat. Commun.*, 2022, 13, 1189; *Nat. Commun.*, 2022, 13, 118; *Nat. Commun.*, 2021, 12, 3502; *J. Am. Chem. Soc.*, 2022, 144, 6028; *Angew. Chem. Int. Ed.* 2021, 60, 16622), the exact mechanism is unclear, especially in neutral media. There still lacks direct experimental evidence to show the hydrogen spillover effect towards promoting HER.

The novelty and significance of this work is re-emphasized as the following:

(1) HER can take place in acidic, neutral and alkaline medium. Compared with acidic HER, HER in neutral media is much more difficult, which is mainly limited by the low concentration of protons in neutral electrolytes, and water dissociation is needed to produce H species. The water dissociation process has to overcome a high energy barrier of about 1.0 eV and thus it becomes the rate-determining step for HER in neutral media, especially at large overpotentials. To resolve this tough issue, we propose a new catalytic structure with the combination of oxygen-deficient WO_{3-x} and Ru nanoparticles, where WO_{3-x} has the ability to enrich and concentrate protons from the electrolyte as well as store protons, and Ru has optimum hydrogen adsorption energy for hydrogen recombination. Specifically, protons in the electrolyte can be enriched and concentrated at WO_{3-x} by an insertion process. Then, the inserted protons in WO_{3-x} can spillover to Ru to facilitate the HER. As the further increase of overpotential, WO_{3-x} with the low water dissociation energy of 0.35 eV can dissociate water to generate protons, which can also spillover to Ru. Therefore, the hydrogen coverage on Ru is significantly increased and thus the Ru- WO_{3-x} catalyst exhibits outstanding HER activity in neutral media. Therefore, the delicate integration of two components with their pre-designed functions has been demonstrated to be highly efficient in increasing the hydrogen coverage at the catalytic sites and hence boosting HER activity in neutral media.

(2) Reversible hydrogen spillover in HER and the detailed clarity of H transfer from oxide support to noble metal nanoparticles are reported for the first time. Typically, hydrogen spills over from precious metal to oxide in heterogeneous thermal catalysis. Recently, this conventional H transfer phenomenon has also been found in HER, but all of them are in acidic media (*Nat. Commun.*, 2022, 13, 1189; *Nat. Commun.*, 2022, 13, 118; *J. Am. Chem. Soc.*, 2022, 144, 6028; *Angew. Chem. Int. Ed.*, 2021, 60, 16622). In acidic media, the H transfer is believed from noble metal to the support as noble metals directly get protons from acidic electrolyte owing to their optimum hydrogen

adsorption ability, thus resulting in a high hydrogen coverage on the noble metal surface. In this case, the support mainly receives H atoms transferred from noble metal and provides a place for hydrogen recombination. Furthermore, most reports about the H transfer process lack of solid evidences, especially the direct confirmation such as *in situ* characterizations; the H transfer process is mostly deduced from experimental phenomena. Our work is the first one demonstrating **reversible H spillover from oxide support to noble metal** for HER, especially in neutral media. By combining *in situ* Raman spectroscopy, *operando* EIS characterizations and DFT calculations, we unraveled this H transfer from oxygen-deficient WO_{3-x} to Ru nanoparticles in detail and **for the first time** directly observed this process. The finding of this reversible H transfer provides a strong theoretical guide for the design of highly active HER catalysts in neutral media, which shall greatly open up the design ideas for other electrochemical reactions such as electrochemical hydrogenation.

The term spillover was introduced in the sixties to explain just the migration of hydrogen atoms from platinum to WO_3 used as support. Since the nineties there is a plenty of works ... it was a controversial topic because of the lack of a scientific basis for discussion. Recently, there was a revival. There is an excellent review (not cited by the authors) by R. Prins (Chem. Rev. 2012, 112, 2714–2738) where the facts and fiction are discussed.

Response: We thank the reviewer for the comment and suggestion. We fully understand the reviewer's concern here. We have carefully read and cited the review paper mentioned by the reviewer (Ref. 24 in the manuscript; Page 13, Line 37):

The review paper first summarizes spillover on non-reducible supports, semiconductors and defective insulators, and then discusses spillover for hydrogen storage, methanol synthesis, hydroisomerization, and hydrogenation reaction. The authors declared that spillover of H atoms from metal particles (or other sources of H atoms) to a reducible support is a fact, which was demonstrated by Khoobiar in the first publication on spillover. In this review paper, the authors emphasized that spillover from a metal surface to the surface of a defect-free support such as Al_2O_3 , SiO_2 , MgO , and zeolites is energetically impossible and the H-D exchange experiment on the support is not the proof of spillover in these systems, and direct solid proof for the spillover on the defect-free support is still lacking so far. The review paper only discussed spillover in the thermal catalytic reactions including hydrogen storage, methanol synthesis, hydrogenation and hydroisomerization. Overall, the review paper emphasizes that spillover from a metal surface to the surface of a defect-free support needs to be well documented, not just an H-D exchange experiment.

In this work, by combining *in situ* Raman spectroscopy and *operando* EIS characterizations, we unraveled reversible hydrogen spillover from oxygen-deficient WO_{3-x} to Ru nanoparticles in detail and **for the first time** directly observed this process. As a supplementary tool, DFT calculation has been widely used as a convincing means to study reaction pathways, rate-limiting steps, and reaction

intermediates. Here, DFT calculation with reasonable models and detailed analysis reveals the potential reaction mechanism involving migration of H atoms from WO_{3-x} to the surface of Ru. Furthermore, the Tafel slope of Ru-WO_{3-x}/CP decreases to 41 mV dec⁻¹ compared to that of Ru/C (5 wt.%)/CP (78 mV dec⁻¹). This indicates that the rate-limiting step changes from water dissociation on Ru/C to hydrogen recombination on Ru-WO_{3-x}/CP. This result experimentally supports the reaction mechanism revealed by the DFT calculations.

Therefore, I don't think, this work is suitable for publication in Nature Communications.

Response: We highly appreciate the reviewer's constructive comments and suggestions, based on which, our point-by-point responses are listed below.

Second, although the exhaustive analysis performed by the authors, I do not coincide with the interpretations. According to a reference mentioned by the authors (Angew. Chem., Int. Ed. 58, 16038-16042 (2019)), the tendency is the migration of hydrogen atoms from Pt to WO_(3-x). Why should go in the opposite direction as proposed in this work?

Response: We thank the reviewer for raising the question. In the cited work (Angew. Chem., Int. Ed. 58, 16038-16042 (2019)), the proposed migration of hydrogen atoms is from Pt to WO_{3-x}, which occurs in 0.5 M H₂SO₄ electrolyte. Since protons in acidic electrolyte are abundant and Pt has a more favorable hydrogen adsorption than WO_{3-x}, the hydrogen coverage on Pt is anticipated to be more than that on WO_{3-x}. Additionally, based on the electrochemical characterizations, the authors attributed the enhanced HER performance to the hydrogen spillover from Pt to WO_{3-x} (**no direct experimental evidence was provided to support this hydrogen spillover**).

In our work, HER took place in neutral reaction media (1.0 M PBS), the concentration of protons in the electrolyte is very low. In this case, water has to undergo dissociation to release protons for the subsequent proton adsorption and hydrogen recombination on catalyst's surface. The HER in neutral medium is typically rate-limited at the water dissociation step.

For the Ru-WO_{3-x} catalytic system, protons can be directly inserted into WO_{3-x} to form H_yWO_{3-x}. The intercalation process of protons in WO_{3-x} can be written as:

This process occurs at a potential more negative to the peak of proton insertion into WO_{3-x}. And protons can be easily inserted into the hexagonal tunnels of WO_{3-x} and then coordinated with terminal oxygen atoms (*ACS Appl. Mater. Interfaces*, 2014, 6, 18901-18910). At the same time, the electrons injected from the electrode reduce the adjacent W atom from W⁶⁺ to W⁵⁺ (*ACS Appl. Mater. Interfaces*, 2016, 8, 13966-13972).

The proton insertion peak of Ru-WO_{3-x} in 1.0 M PBS is at 0.288 V vs. RHE (Fig. 4i), which is much more positive than the theoretical onset potential for proton adsorption over Ru for HER. This reveals that protons in the electrolyte can be inserted into WO_{3-x} under the negative applied potentials. Then, the inserted protons

in WO_{3-x} can spillover to Ru to facilitate the HER. As the further increase of overpotential, WO_{3-x} will dissociate the water to generate protons, which can also spillover to Ru. Therefore, hydrogen atoms tend to migrate from WO_{3-x} to Ru.

More importantly, by combining *in situ* Raman spectroscopy and *operando* EIS characterizations, we unraveled this reversible hydrogen spillover from oxygen-deficient WO_{3-x} to Ru nanoparticles in detail and **for the first time** directly observed this process.

Figure 4i. CV curve of Ru- WO_{3-x} recorded in 1.0 M PBS. Scan rate: 5 mV/s.

Where does the hydrogen in WO_{3-x} come from? the dissociation of water (according to the calculations of the authors) is about 0.87 eV!! the value of this barrier for the dissociation on Ru cluster is not given.

Response: The hydrogen in WO_{3-x} comes from two sources. The first one is from the protons in the electrolyte, which can be inserted into WO_{3-x} under negative potential. Another is from the dissociation of water on WO_{3-x} at large overpotentials. As revealed in Supplementary Figs. 22-24, the barrier for water dissociation on Ru is about 0.46 eV and that on WO_{3-x} is around 0.35 eV. The water dissociation on WO_{3-x} is more favorable than that on Ru.

For the Ru- WO_{3-x} catalytic system, protons can be directly inserted into WO_{3-x} to form $\text{H}_y\text{WO}_{3-x}$. The intercalation process of protons in WO_{3-x} can be written as:

This process occurs at a potential more negative to the peak of proton insertion into WO_{3-x} . And protons can be easily inserted into the hexagonal tunnels of WO_{3-x} and then coordinated with terminal oxygen atoms (*ACS Appl. Mater. Interfaces*, 2014, 6, 18901-18910). At the same time, the electrons injected from the electrode reduce the adjacent W atom from W^{6+} to W^{5+} (*ACS Appl. Mater. Interfaces*, 2016, 8, 13966-13972).

The proton insertion peak of Ru- WO_{3-x} in 1.0 M PBS is at 0.288 V vs. RHE (Fig. 4i), which is much more positive than the theoretical onset potential for proton adsorption over Ru for HER. This reveals that protons in the electrolyte can be inserted into WO_{3-x} under the negative applied potentials. Then, the inserted protons in WO_{3-x} can spillover to Ru to facilitate the HER. As the further increase of overpotential, WO_{3-x} will dissociate the water to generate protons, which can also spillover to Ru.

Moreover, to further explore the reaction process, we calculated the energy barriers

of the Tafel step and Heyrovsky step on Ru. As shown in Supplementary Figs. 27-29, the Heyrovsky step shows a lower energy barrier than the Tafel step, suggesting that the hydrogen atoms on Ru transferred from WO_{3-x} tend to follow the Heyrovsky mechanism to generate di-hydrogen. As the energy barrier of hydrogen transfer from WO_{3-x} to Ru is only 0.25 eV, the whole process to generate di-hydrogen including hydrogen atoms spillover from WO_{3-x} to Ru followed by the Heyrovsky step to generate di-hydrogen has an energy barrier of 0.35 eV, indicating Heyrovsky step is the rate limiting step in this process. However, the di-hydrogen produced by water dissociation over Ru itself has a higher energy barrier of 0.46 eV, suggesting that this process is more difficult than the spillover mechanism to generate molecular hydrogen.

Supplementary Figure 22. Energy barrier of water dissociation process on Ru of $\text{Ru-H}_x\text{WO}_{3-x}$ (a) and WO_{3-x} of $\text{Ru-H}_x\text{WO}_{3-x}$ (b). (TS represent the transition states).

Supplementary Figure 23. Side and top view illustrations of DFT models used for calculating the energy barrier of water dissociation process on Ru of $\text{Ru-H}_x\text{WO}_{3-x}$. (a) Initial state. (b) Transition state. (c) Final state. The blue, red and purple balls represent W, O and Ru atoms. The green and yellow balls represent the H atoms inserted into WO_{3-x} and the H atoms involved in HER.

Supplementary Figure 24. Side and top view illustrations of DFT models used for calculating the energy barrier of water dissociation process on WO_{3-x} of $\text{Ru-H}_x\text{WO}_{3-x}$. (a) Initial state. (b) Transition state. (c) Final state. The blue, red and purple balls represent W, O and Ru atoms. The green and yellow balls represent the H atoms inserted into WO_{3-x} and the H atoms involved in HER.

Supplementary Figure 27. Energy barrier of Heyrovsky step (a) and Tafel step (b) on Ru of $\text{Ru-H}_x\text{WO}_{3-x}$. (TS represent the transition states).

Supplementary Figure 28. Side and top view illustrations of DFT models used for calculating the energy barrier of Heyrovsky step on Ru clusters of $\text{Ru-H}_x\text{WO}_{3-x}$. (a) Initial state. (b) Transition state. (c) Final state. The blue, red and purple balls represent W, O and Ru atoms. The green and yellow balls represent the H atoms inserted into WO_{3-x} and the H atoms involved in HER.

Supplementary Figure 29. Side and top view illustrations of DFT models used for calculating the energy barrier of Tafel step on Ru clusters of Ru-H_xWO_{3-x}. (a) Initial state. (b) Transition state. (c) Final state. The blue, red and purple balls represent W, O and Ru atoms. The green and yellow balls represent the H atoms inserted into WO_{3-x} and the H atoms involved in HER.

The authors estimate a capacitance enhancement of about 20 for the system WO_(3-x). This means a much larger real area for this system. Could be the enhancement of also about 20 observed for the HER caused by this increase in the area? a larger roughness factor?

Response: We thank the reviewer for raising the question. Firstly, we calculated the roughness factor (R_f) based on the double layer capacitance (C_{dl}) by the following equation:

$$R_f = \frac{C_{dl}}{0.04}$$

where the C_{dl} (F/g) is the double-layer capacitance of catalysts (Supplementary Fig. 5), and 0.04 (mF/cm²) is the general double-layer capacitance for a smooth surface for metal oxides (*Appl. Catal., B*, 2022, 316, 121602.).

The R_f of Ru/C and Ru-WO_{3-x} is calculated to be 100 and 55, respectively. The R_f of Ru-WO_{3-x} is only 55% of Ru/C, indicating that roughness factor should not be a decisive factor in significantly increasing HER activity.

Then, we calculated the electrochemically active surface area (ECSA) based on the R_f and obtained the LSV curves normalized by the ECSA. As displayed in Supplementary Fig. 6, Ru-WO_{3-x}/CP still shows much better HER activity than Ru/C (5 wt.%)/CP and the current density of Ru-WO_{3-x}/CP is enhanced by a factor of 34 as compared to the Ru/C (5 wt.%)/CP at the potential of -0.150 V vs. RHE.

Supplementary Figure 5. (a) CV curves of Ru/C (5 wt.%) recorded at various scan rates. (b) The capacitive current measured at 1.107 V (vs. RHE) plotted as a function of scan rate of Ru/C (5 wt.%). (c) CV curves of Ru-WO_{3-x} recorded at various scan rates. (d) The capacitive current measured at 1.107 V (vs. RHE) plotted as a function of scan rate of Ru-WO_{3-x}.

Supplementary Figure 6. LSV curves normalized by ECSA of Ru-WO_{3-x}/CP and Ru/C (5 wt.%)/CP.

Minors details: the authors give sometimes values of potential vs RHE and sometimes vs Ag/AgCl according to the electrode reference used in each experiment. It should be clearer for the reader if the values are informed always relative to the same reference electrode.

Response: We thank the reviewer for the nice suggestion. Except for the *in-situ* Raman spectroscopy where the potential is relative to Ag/AgCl electrode, the potential elsewhere in the work is relative to RHE. We have explained this in the experimental section to make it clear for the readers (Page 11, Line 44-45).

“In this work, only the potentials in the Raman spectra are relative to the Ag/AgCl electrode; all other potentials are relative to the RHE.”

Reviewer #2 (Remarks to the Author):

The present manuscript by Chen, Liu, Wang and colleagues compares Ru catalysts for the HER reaction in neutral pH water splitting in the presence of hydrogen treated tungsten oxide. A significant increase in activity is observed in the presence of tungsten oxide. The authors ascribe the increase in activity by looking at hydrogen spillover. The authors provide a range of experimental activity measurements and characterization tools. In addition, the authors provide theoretical calculations to explore their experimental findings.

Even though, similar findings have been reported before using Pt instead of Ru (Angew. Chemie Int. Ed. 58, 16038-16042, 2019), I think that the work has the potential to provide essential mechanistic insights to warrant the publication in Nature Communications. However, in the current state the manuscript lacks a correct description of the involved spillover chemistry and the attempted theoretical explanation is not considering the most crucial step – the water dissociation reaction. If this the concept of two active structures (water activation on a metal oxide and combination/desorption on a metal) is convincingly shown, an important mechanistic path for HER is shown.

I suggest the publication of the manuscript, after the authors provided convincing theoretical insights into the different pathways of water dissociation for tungsten oxide and Ru. In addition, they should revise their wording and description of the involved chemistry by considering the existing literature on hydrogen spillover (which is currently not appearing in the manuscript).

Response: We appreciate the reviewer's constructive comments and suggestions, response to which shall greatly improve the quality of our manuscript. Below we give a point to point response to the question/concern raised by the reviewer. In this new version of the manuscript, we have addressed all of the concerns/questions raised by the reviewer on both the experimental and theoretical aspects, updated additional characterizations and analyses accordingly. The water dissociation reaction on both WO_{3-x} and Ru have been calculated and analyzed in detail. Accordingly, the wording and description of the involved spillover chemistry has been carefully revised. Moreover, we have read all the literatures recommended by the reviewer and cited them properly in the manuscript.

This is detailed in the following comments.

Major comments

1. Reaction mechanism: The authors report that the rate limiting process for Ru/C catalysts is water dissociation. They observe a change in the rate limiting step (rls) when adding reduced " WO_{3-x} " to the catalyst. Their kinetic modelling and in situ Raman spectroscopy suggest that H combination and desorption is the rls for this catalyst under their testing conditions. Now, the authors describe " WO_{3-x} " as a proton storage medium, which increases the H coverage of Ru. This discussion misses the relevant point: Why is the water dissociation reaction not the rls in the presence of " WO_{3-x} "?

The authors do not provide a satisfying discussion and theoretical evaluation of this

question. To me, it appears that the reduced “WO_{3-x}” is now acting as a catalyst for water dissociation (which is faster than Ru) and Ru mainly acts as H combination catalyst.

Therefore, the theoretical calculation do not discuss the relevant process and provide no explanation why Ru/WO_{3-x} performs better than Ru/C. The calculated spillover process of noble metals to or from tungsten oxide have been calculated and discussed before (see below) for heterogeneous catalysis and the presented theoretical results (H can hop from WO₃ to Ru) do not appear novel. The authors should compare the dissociation of water and a Ru (NP) surface vs. the dissociation on “WO_{3-x}” with theoretical calculation. This would provide significant new insights in the interplay of different active centers.

Response: Thanks for the reviewer’s valuable comment. To clarify the above point, we first calculated the water dissociation energy on Ru cluster and WO_{3-x}, respectively. As revealed in Supplementary Figs. 22-24, the barrier for water dissociation on Ru cluster is 0.46 eV and that on WO_{3-x} is 0.35 eV. The water dissociation on WO_{3-x} is more favorable than that on Ru cluster. This means WO_{3-x} can generate protons by water dissociation occurring on itself at large overpotentials.

Moreover, to explore the H recombination process after hydrogen spillover from WO_{3-x} to Ru clusters, we calculated the energy barriers of Tafel step and Heyrovsky step on Ru clusters. As shown in Supplementary Figs. 27-29, the Heyrovsky step shows a lower energy barrier than the Tafel step, suggesting that the hydrogen atoms on Ru transferred from WO_{3-x} tend to follow the Heyrovsky mechanism to generate di-hydrogen. As the energy barrier of hydrogen transfer from WO_{3-x} to Ru is only 0.25 eV, the whole process to generate di-hydrogen including hydrogen atoms spillover from WO_{3-x} to Ru followed by the Heyrovsky step to generate di-hydrogen has an energy barrier of 0.35 eV, indicating Heyrovsky step is the rate limiting step in this process. However, the di-hydrogen produced by water dissociation over Ru itself has a higher energy barrier of 0.46 eV, suggesting that this process is more difficult than the spillover mechanism to generate molecular hydrogen.

Moreover, we would like to describe the mechanism of HER proposed in this work: At low overpotentials, the water dissociation on both WO_{3-x} and Ru is sluggish and insufficient to provide enough protons for HER. In this case, WO_{3-x} can capture protons in 1.0 M PBS as the H_yWO_{3-x}. For the Ru-WO_{3-x} catalytic system, protons can be directly inserted into WO_{3-x} to form H_yWO_{3-x}. The intercalation process of protons in WO_{3-x} can be written as:

This process occurs at a potential more negative to the peak of proton insertion into WO_{3-x}. And protons can be easily inserted into the hexagonal tunnels of WO_{3-x} and then coordinated with terminal oxygen atoms (*ACS Appl. Mater. Interfaces*, 2014, 6, 18901-18910). At the same time, the electrons injected from the electrode reduce the adjacent W atom from W⁶⁺ to W⁵⁺ (*ACS Appl. Mater. Interfaces*, 2016, 8, 13966-13972). The proton insertion peak of Ru-WO_{3-x} in 1.0 M PBS is at 0.288 V vs. RHE (Fig. 4i), which is much more positive than the theoretical onset potential for proton adsorption over Ru for HER. This reveals that protons in the electrolyte can be

inserted into WO_{3-x} under the negative applied potentials. Then, the inserted protons in WO_{3-x} can spillover to Ru to facilitate the HER. With further increase of the overpotential, the applied potential is enough to drive water dissociation. In this period, WO_{3-x} starts to dissociate the water to generate hydrogen atoms and they can also spillover to Ru. Therefore, the protons directly captured from electrolyte and generated by water dissociation provide hydrogen atoms for spillover process in this stage.

Supplementary Figure 22. Energy barrier of water dissociation process on Ru of $\text{Ru-H}_x\text{WO}_{3-x}$ (a) and WO_{3-x} of $\text{Ru-H}_x\text{WO}_{3-x}$ (b). (TS represent the transition states).

Supplementary Figure 23. Side and top view illustrations of DFT models used for calculating the energy barrier of water dissociation process on Ru of $\text{Ru-H}_x\text{WO}_{3-x}$. (a) Initial state. (b) Transition state. (c) Final state. The blue, red and purple balls represent W, O and Ru atoms. The green and yellow balls represent the H atoms inserted into WO_{3-x} and the H atoms involved in HER.

Supplementary Figure 24. Side and top view illustrations of DFT models used for calculating the energy barrier of water dissociation process on WO_{3-x} of $\text{Ru-H}_x\text{WO}_{3-x}$. (a) Initial state. (b) Transition state. (c) Final state. The blue, red and purple balls represent W, O and Ru atoms. The green and yellow balls represent the H atoms inserted into WO_{3-x} and the H atoms involved in HER.

Supplementary Figure 27. Energy barrier of Heyrovsky step (a) and Tafel step (b) on Ru of $\text{Ru-H}_x\text{WO}_{3-x}$. (TS represent the transition states).

Supplementary Figure 28. Side and top view illustrations of DFT models used for calculating the energy barrier of Heyrovsky step on Ru clusters of $\text{Ru-H}_x\text{WO}_{3-x}$. (a) Initial state. (b) Transition state. (c) Final state. The blue, red and purple balls represent W, O and Ru atoms. The green and yellow balls represent the H atoms inserted into WO_{3-x} and the H atoms involved in HER.

Supplementary Figure 29. Side and top view illustrations of DFT models used for calculating the energy barrier of Tafel step on Ru clusters of Ru-H_xWO_{3-x}. (a) Initial state. (b) Transition state. (c) Final state. The blue, red and purple balls represent W, O and Ru atoms. The green and yellow balls represent the H atoms inserted into WO_{3-x} and the H atoms involved in HER.

Figure 4i. CV curve of Ru-WO_{3-x} recorded in 1.0 M PBS. Scan rate: 5 mV/s.

2. Discussion of spillover: Noble metal/WO₃ catalysts is the archetypical system for hydrogen spillover (first reported 1964). In the classical definition hydrogen spillover describes the dissociative adsorption of di-hydrogen on a (metal)-surface and the subsequent spillover to a surface (often a metal oxide) which is kinetically not able to adsorb di-hydrogen on its own. The reversible process (hydrogen combines on a metal surface and desorbs) is generally called reversible spillover. The authors describe this process here. Consequently, the discussion should be adopted to this.

In this respect, the authors should consider literature which currently appears to be not considered. The field of heterogeneous (thermal) catalysis provides an extensive report on spillover, amongst others:

- Khoobiar, S. Particle to Particle Migration of Hydrogen Atoms on Platinum—Alumina Catalysts from Particle to Neighboring Particles. *J. Phys. Chem.* 68, 411–412 (1964).
- Miu, E. V. & McKone, J. R. Comparisons of WO₃ reduction to H_xWO₃ under thermochemical and electrochemical control. *J. Mater. Chem. A* 7, 23756–23761 (2019).
- Prins, R. Hydrogen Spillover. Facts and Fiction. *Chem. Rev.* 112, 2714–2738 (2012).
- Levy, R. & Boudart, M. The kinetics and mechanism of spillover. *J. Catal.* 32, 304–314 (1974).
- Boudart, M. 'On the nature of spilt-over hydrogen'. *J. Mol. Catal. A Chem.* 138, 319–321 (1999).

Also, I recommend the change of the title. This is by no means “unconventional”, a more suitable title appears to be “Reversible hydrogen spillover from WO_{3-x} to Ru in Ru-WO_{3-x} enhances hydrogen evolution activity in neutral pH water splitting”.

Response: We thank the reviewer for the valuable comments and suggestions. According to these comments and suggestions, we have revised the hydrogen spillover to the reversible hydrogen spillover in this work and modified the title of the

work accordingly (Page 1, Line 1). Moreover, we have carefully read and cited the references recommended by the reviewer (Page 13, Line 33-40).

“22. Khoobiar, S. Particle to particle migration of hydrogen atoms on platinum-alumina catalysts from particle to neighboring particles. *J. Phys. Chem.* 68, 411-412 (1964).

23. Miu, E. V., McKone, J. R. Comparisons of WO_3 reduction to H_xWO_3 under thermochemical and electrochemical control. *J. Mater. Chem. A* 7, 23756-23761 (2019).

24. Prins, R. Hydrogen spillover. Facts and fiction. *Chem. Rev.* 112, 2714-2738 (2012).

25. Levy, R., Boudart, M. The kinetics and mechanism of spillover. *J. Catal.* 32, 304-314 (1974).

26. Roland, U., Braunschweig, T., Roessner, F. On the nature of spilt-over hydrogen. *J. Mol. Catal. A Chem.* 127, 61-84 (1997).”

3. Discussion of the tungsten oxide structure: Hydrogen treatments of WO_3 as know to produce H_xWO_3 bronzes (see literature). This is especially true for WO_3 in contact with a noble metal. The authors do a hydrogen pretreatment and consequently describe the materials “oxygen-deficient WO_{3-x} ”, this appears to be an assumption that is not justified by the authors experiments. EPR (line 109-111) and XPS (line 116-121) is indicative that W(VI) is transformed to W(V) but do not provide the evidence an evidence that indeed vacancies are formed. Hydrogen treatment will also form the H_xWO_3 bronze which will lead to a similar observed reduction of W. Therefore, the authors should amend their discussion to this fact and should consider to use $\text{H}_x\text{WO}_{3-y}$ instead of WO_{3-x} throughout their manuscript.

Response: Thanks for the reviewer’s valuable comments. As the reviewer described, hydrogen treatment can form H_xWO_3 bronze which will lead to a similar observed reduction of W. This leads to the appearance of W(V) in XPS. Therefore, we have revised the corresponding discussion in the XPS analysis (**Page 4, Line 26-29**).

“Additionally, two more deconvoluted peaks at 34.53 eV and 36.63 eV can be assigned to W(V)^{43, 45}. Notably, a few hydrogen atoms may be induced in the surface of WO_{3-x} support in the H_2/Ar reduction process and this may also lead to the appearance of W(V).”

EPR is able to distinguish the signal of W(V) and oxygen vacancies: the signal at $g = 2.004$ is associated with unpaired electrons trapped by oxygen vacancies rather than by W(V) ($g = 1.89, 1.94$ or 2.19) (*Ceram. Int.*, 2022, 48, 4115-4123; *Appl. Catal., B*, 2018, 239 398-407; and *RSC Adv.*, 2017, 7, 2351). Therefore, the EPR signal at $g = 2.004$ reveals the presence of oxygen vacancies in $\text{Ru-H}_x\text{WO}_{3-x}$.

To further investigate oxygen vacancies in Ru-WO_{3-x} , we conducted O_2 -temperature programmed desorption (O_2 -TPD). Generally, the desorption peak at low temperature ($< 200^\circ\text{C}$) is associated to desorption of physically adsorbed oxygen species. The desorption peaks at 200-600 $^\circ\text{C}$ correspond to desorption of chemisorbed oxygen species, and the peak above 700 $^\circ\text{C}$ is attributed to the release of lattice oxygen (*Catal. Sci. Technol.*, 2016, 6, 3845; *RSC Adv.*, 2019, 9, 7723). As shown in

Supplementary Fig. 4, compared with WO_3 , Ru-WO_{3-x} and WO_{3-x} present shifts of desorption peaks of chemisorbed oxygen species to lower temperatures with much higher intensities of desorption peaks at around 400 °C, which should be related to the chemically adsorbed oxygen adjacent to oxygen vacancies. This result reveals oxygen vacancies in WO_{3-x} and Ru-WO_{3-x} , which matches well with the EPR and O 1s XPS results.

Additionally, if hydrogen inserted into the bulk structure of WO_{3-x} reaches a certain amount and thus forms tungsten bronzes, the $\text{Ru-H}_x\text{WO}_{3-x}$ will show different diffraction peaks with that of Ru-WO_{3-x} (*J. Mater. Chem. A*, 2018, 6, 6780-6784). From the XRD of $\text{Ru-WO}_{3-x}/\text{CP}$ (Fig. 2b), it can be seen that Ru-WO_{3-x} still keeps the original crystalline structure of hexagonal WO_3 . This means that the bulk structure is still the hexagonal WO_3 . Therefore, based on the above characterizations, we feel it is reasonable to use WO_{3-x} instead of $\text{H}_x\text{WO}_{3-x}$ to represent the hydrogen treated WO_3 .

Supplementary Figure 4. O_2 -TPD profiles of Ru-WO_{3-x} , WO_{3-x} and WO_3 .

Oxygen temperature programmed desorption (O_2 -TPD) analysis was performed on a BELCAT II fully automatic chemisorber instrument (MicrotracBEL). The procedures were as follows: (1) each sample was pretreated under a He flow (50 mL min^{-1}) at 300 °C for 30 min; (2) the sample was purged with 5% O_2/He for 1 h at 50 °C for O_2 adsorption; (3) the sample was heated to 700 °C at a heating rate of 10 °C min^{-1} under a pure He gas flow. The signal of O_2 desorption was measured by a thermal conductivity detector.

Fig. 2b XRD patterns of WO₃/CP and Ru-WO_{3-x}/CP

Minor comments

- Theoretical calculations: What is the final state of the calculated H₂? Gas phase or in solution? The desorption/adsorption in liquid media is highly complex and is therefore hard to model (e.g. *Angew. Chemie Int. Ed.* 58, 3527–3532, 2019). The authors should be specific which final state of H₂ was assumed/calculated.

Response: Thanks for the reviewer’s valuable comment. The final state of the calculated H₂ is in gas phase and we have added this statement in the DFT calculation description of the experimental section (**Page 12, Line 29-30**).

“In this work, the final state of the calculated H₂ is in gas phase.”

- line 70: “we need to find ways to increase θH.” This seems not correct; increasing the coverage is the consequence of enhancing the water dissociation reaction, but should not be the goal. I suggest to revise the sentence.

Response: We thank the reviewer for the nice suggestion. Accordingly, we have revised this sentence (**Page 3, Line 3-4**).

“Therefore, effective strategies need to be proposed to enhance the HER activity of Ru/C in neutral media.”

- line 120: “In O1s XPS spectrum, two deconvoluted peaks centered at 531.70 eV and 530.30 eV can be assigned to oxygen vacancy and lattice oxygen, respectively.” Even though, such assignment is reported, such assignment is incorrect. The XPS signal of the O1s originates from the electrons of an oxygen atom. An oxygen vacancy has no oxygen atom from which the signal can come from. Therefore, this second peak can have other origins: The presence of OH groups or a lattice oxygen bound to a W(V) atom (close to a vacancy). Both would cause a shift in binding energy compared to lattice oxygen in WO₃. The authors should change the discussion and wording.

Response: We appreciate the reviewer for the nice suggestion. We have changed the discussion of the O 1s XPS spectrum to the following (**Page 4, Line 29-31**):

“In O 1s XPS spectrum, two deconvoluted peaks can be observed. The peak centered at 531.70 eV can be assigned to the OH group or a lattice oxygen bounded to a W(V) atom (close to a vacancy)⁴⁵. Another peak at 530.30 eV is ascribed to the lattice

oxygen.”

- Based on the (S)TEM measurements, the authors should make a statement on the size of the Ru nanoparticles.

Response: Thanks for the reviewer’s valuable suggestion. Accordingly, we have provided the size distribution of Ru nanoparticles in the manuscript and added it in the supplementary information (**Page 4, Line 15-16**).

“The average size of the Ru NPs for Ru-WO_{3-x}/CP is 3.5 nm (Supplementary Fig. 2h).”

Supplementary Figure 2h. Size distribution profiles of Ru NPs for Ru-WO_{3-x}/CP.

- The introduction (line 33) should provide more details on advantages/disadvantages of alkali vs. acidic conditions in water splitting.

Response: Thanks for the reviewer’s valuable comment. We have provided more discussion on the advantages/disadvantages of alkali vs. acidic conditions in water splitting. The details are as follows (**Page 1, Line 33-37**):

“Although HER in acidic condition exhibits better activity, equipment and catalyst corrosion limit the lifetime of operation. Neutral media provides a favorable condition for catalysts to remain stable and less corrosive environment for electrolyzers⁹. And electrolyzers capable of operating in neutral media offer the possibility of achieving hydrogen production directly from seawater without the need for desalination^{9,10}.”

- Line 26: “to meet the energy dilemma” It is not clear what the “energy dilemma” is. I suggest rephrasing.

Response: Thanks for the reviewer’s comment. We have revised the sentence accordingly (**Page 1, Line 26-27**).

“Hydrogen, with high gravimetric energy density, is an ideal candidate to replace the traditional fossil fuels”

- Figure 4b: Raman vibration frequency should be labeled with Ru-D (or Ru-2H) not Ru-H.

Response: Thanks for the reviewer’s comment. We have revised this typo accordingly.

Fig. 4b *In-situ* Raman spectra of Ru-WO_{3-x}/CP recorded from -0.2 to -0.7 V vs. Ag/AgCl in 1.0 M PBS (in D₂O).

- The materials and method section is reporting the synthesis protocols for Pt, Ni-WO₃ and Ru-MoO₃ catalysts, which were not discussed in the manuscript. The performance of these materials is clearly different but may be presented in another manuscript. The authors should remove this section or discuss the data in the manuscript.

Response: Thanks for the reviewer's comment. Accordingly, we have removed this section.

- Line 309: This is not the Discussion section; this is the conclusion/summary. I suggest to remove the heading or change to conclusion. The authors discuss along with the presentation of the results. The section (line 57) should be labelled "Results and discussion")

Response: Thanks for the reviewer's comment. Accordingly, we have revised accordingly (**Page 2, Line 15; Page 10, Line 14**).

Reviewer #3 (Remarks to the Author):

In this work, Chen et al. integrated Ru NPs on oxygen-deficient WO_{3-x} for HER in neutral electrolyte, and proposed that protons inserted into WO_{3-x} can be transferred to Ru NPs during HER. The obtained Ru- WO_{3-x} enhances HER activity by a factor of 24.0 as compared to the commercial Ru/C. They also indicated the hydrogen spillover phenomenon through combined in situ Raman spectroscopy investigations, electrochemical measurements and DFT calculations. Overall, this work explained the hydrogen spillover mechanism in detail, however, there are a list of issues that should firstly be addressed prior to consideration for publication in Nat. Commun.

Response: We appreciate the reviewer's constructive comments and suggestions, response to which shall greatly improve the quality of our manuscript. Below we give a point to point response to the question/concern raised by the reviewer.

1. The latest research progress of H spillover and application prospect of neutral HER should be discussed and added in the Introduction.

Response: We thank the reviewer for the nice suggestion. Accordingly, we have carefully revised the Introduction of the manuscript (**Page 2, Line 7-13**).

"Hydrogen spillover, the migration of activated hydrogen atoms generated by dissociation of di-hydrogen adsorbed on a metal surface into a reducible metal oxide support, is a common phenomenon in heterogeneous catalysis²²⁻²⁶. Recently, hydrogen spillover strategy has been taken into account for the catalyst design to achieve compelling HER performance, such as Pt alloys-CoP²⁷, Pt/CoP²⁸, and Pt/TiO₂²⁹ electrocatalysts. However, spillover strategies have rarely been studied on neutral HER and the exact mechanism of hydrogen spillover to improve HER is still unclear."

2. The XRD peaks of Ru- WO_{3-x} in Fig. 2b show obvious shift to higher angles, while the authors did not explain it.

Response: Thanks for the reviewer's comment. The shift of Ru- WO_{3-x} in XRD is caused by oxygen vacancies, which leads to lattice shrinkage. We added this explanation into the revised manuscript (**Page 4, Line 1-2**).

"The Ru- WO_{3-x} /CP displays clear diffraction peaks of hexagonal WO_3 (JCPDS No. 85-2460)⁴², but no diffraction peaks related to Ru NPs, possibly due to their small sizes and low content. In addition, the peaks of Ru- WO_{3-x} /CP are shifted to high angles, indicating lattice shrinkage, which may be caused by oxygen vacancies⁴⁰."

3. The WO_{3-x} reference sample should be well characterized, especially for oxygen vacancies.

Response: Thanks for the reviewer's nice suggestion. According to the reviewer's suggestion, we have provided the XPS spectra of WO_{3-x} /CP. As shown in Supplementary Fig. 3, the WO_{3-x} /CP also displays plenty of oxygen vacancies. To further investigate the oxygen vacancies in WO_{3-x} , we conducted O₂-temperature programmed desorption (O₂-TPD). Generally, the desorption peak at low temperature (< 200°C) is associated to desorption of physically adsorbed oxygen species. The

desorption peaks at 200-600 °C correspond to desorption of chemisorbed oxygen species, and the peak above 700 °C is attributed to the release of lattice oxygen (*Catal. Sci. Technol.*, 2016, 6, 3845; *RSC Adv.*, 2019, 9, 7723). As shown in Supplementary Fig. 4, compared with WO₃, Ru-WO_{3-x} and WO_{3-x} present shifts of desorption peaks of chemisorbed oxygen species to lower temperatures with much higher intensities of desorption peaks at around 400 °C, which should be related to the chemically adsorbed oxygen adjacent to oxygen vacancies. This result reveals oxygen vacancies in WO_{3-x} and Ru-WO_{3-x}, which matches well with the EPR and O 1s XPS results.

Supplementary Figure 3. XPS spectra of WO_{3-x}/CP and Ru-WO_{3-x}/CP. (a) W 4f spectra. (b) O 1s spectra.

Supplementary Figure 4. O₂-TPD profiles of Ru-WO_{3-x}, WO_{3-x} and WO₃.

Oxygen temperature programmed desorption (O₂-TPD) analysis was performed on a BELCAT II fully automatic chemisorber instrument (MicrotracBEL). The procedures were as follows: (1) each sample was pretreated under a He flow (50 mL min⁻¹) at 300 °C for 30 min; (2) the sample was purged with 5% O₂/He for 1 h at 50 °C for O₂ adsorption; (3) the sample was heated to 700 °C at a heating rate of 10 °C min⁻¹ under a pure He gas flow. The signal of O₂ desorption was measured by a thermal conductivity detector.

4. In Fig.4d, the Raman peak intensity of Ru-H firstly remained steady until -0.5 V (vs. Ag/AgCl) and then gradually increased when more positive potentials were applied. The authors explained that it was due to the increase of hydrogen coverage on Ru.

However, Fig. 4f shows that hydrogen coverage increases with the enhancing overpotentials. This is contradictory, please explain.

Response: Thanks for the reviewer's comments. In Fig. 4d, the potential changes from negative to positive. During this period, HER rate slowed down and H consumed on Ru decreased, but the hydrogen coverage on Ru still increased due to the inserted proton spillover from WO_{3-x} into Ru. In Fig. 4f, the applied potential changes from positive to negative. As the overpotential increased, water dissociation occurred on both WO_{3-x} and Ru. During this period, more protons could be generated on the WO_{3-x} surface and then spillover onto Ru, in the meanwhile Ru could also adsorb H through its own water dissociation pathway. The above two factors increased the hydrogen coverage on Ru.

Fig. 4 **d** *In situ* Raman spectra of Ru- WO_{3-x} /CP recorded in 1.0 M PBS from -0.6 to -0.2 V vs. Ag/AgCl. **f** Fitted data of C_ϕ at different overpotentials for various electrocatalysts during HER in 1.0 M PBS.

5. The deuterium isotopic substitution experiment should also be performed on WO_{3-x} .

Response: Thanks for the reviewer's valuable suggestion. Accordingly, we have performed the deuterium isotopic substitution experiment on WO_{3-x} /CP. As shown in Supplementary Fig. 16, there is no obvious difference between the Raman results in deuterium isotopic substitution and non-deuterium isotopic substitution experiments. Only W-O Raman peaks can be seen in the Raman spectra.

Supplementary Figure 16. *In-situ* Raman analysis of WO_{3-x} /CP in deuterium isotopic substitution experiment. *In-situ* Raman spectra of WO_{3-x} /CP recorded in 1.0 M PBS from -0.5 to -0.9 V vs. Ag/AgCl (a) and sweep back from -0.6 to -0.1 V vs. Ag/AgCl (b).

6. The illustration of operando Raman spectroscopy setup should be provided in SI.

Response: Thanks for the reviewer’s suggestion. Accordingly, we have provided the *operando* Raman spectroscopy setup in the SI (Supplementary Fig. 30) and the corresponding description is included in the experimental section (Page 12, Line 11-16).

“The electrochemical cell used for Raman measurement was homemade by Teflon and a quartz plate was employed as the window to cross the laser. A Pt wire and a Ag/AgCl electrode (1.0 M KCl as inner filling electrolyte) were applied as the counter electrode and the reference electrode, respectively. To apply a controlled potential on the catalyst during the Raman measurement, chronoamperometry was performed at various potentials in 1.0 M PBS.”

Supplementary Figure 30. The illustration of the *in-situ* Raman spectroscopy setup.

7. The Tafel slopes were analyzed at different potential regions for three samples. Noted that Tafel slopes show differences in low- and high-potential regions (see J. Am. Chem. Soc. 2022, 144, 6028–6039).

Response: Thanks for the reviewer’s comment. Generally, Tafel slopes show differences in low and high potential regions. Notably, for a specific catalyst, the Tafel slope in low-overpotential region can be used to deduce the HER mechanism since the Faradaic current is small and mass transfer is sufficient for HER in this stage. In the large-overpotential region, the Faradaic current is large and mass transfer will influence the reaction rate. For different catalysts, the onset potentials of them are significantly different. Considering that the Tafel slope should be calculated in the low overpotential region, the Tafel slopes of them are calculated in different potentials. Since $\text{WO}_{3-x}/\text{CP}$ shows a large onset potential, the potential for calculating the Tafel slope is more negative than that of the other catalysts.

8. $\text{Ru-WO}_{3-x}/\text{CP}$ reaches a j of 10 mA cm^{-2} at an overpotential of 19 mV. Although it is superb, ampere-level current densities with relatively low cell voltages should be more meaningful. The paper (J. Am. Chem. Soc. 2022, 144, 6028–6039) proposed that hydrogen spillover-bridged Volmer/Tafel processes can enable 3.5 A cm^{-2} under an overpotential of $251 \pm 3 \text{ mV}$ in 1.0 M KOH electrolyte. The HER activity and stability test of Ru-WO_{3-x} under higher current densities should be conducted to study the effect of hydrogen spillover in this system.

Response: We thank the reviewer for the constructive comments and suggestions. According to these comments and suggestions, we have evaluated the HER activity

and stability under higher current densities. As shown in Supplementary Fig. 7, Ru-WO_{3-x}/CP displays a low overpotential of 225 mV to achieve a current density of 1 A cm⁻², and the potential of Ru-WO_{3-x}/CP remains stable to attain 1 A cm⁻² in the chronopotentiometry test.

Supplementary Figure 7. HER activity and stability test of Ru-WO_{3-x}/CP in 1.0 M PBS. (a) LSV curve of Ru-WO_{3-x}/CP recorded in 1.0 M PBS. (b) Chronopotentiometric stability of Ru-WO_{3-x}/CP under constant current density of 1 A cm⁻² in 1.0 M PBS.

9. Considering the application of alkaline electrolyser, how about the performance of Ru-WO_{3-x} in alkaline electrolyte?

Response: We thank the reviewer for raising the question. WO₃ or WO_{3-x} gradually dissolves in alkaline electrolyte (*ChemCatChem*, 2021, 13, 3836-3845; *Sol. Energ. Mat. Sol. C*, 2020, 207, 110337). Thus, it is not appropriate to apply Ru-WO_{3-x} in alkaline condition because of the stability issue.

10. Could the limiting steps of the work, as suggested through simulations, be supported through the determination of kinetic isotope effects using D₂O?

Response: Thanks for the reviewer's nice suggestion. Since the mobility of deuterium in the D₂O electrolyte is estimated to be 1.6 to 5.0 times slower than the protons in a H₂O solution, D₂O has been demonstrated to be useful to study the influence of proton transfer kinetics in electrochemical reactions (*Chinese J. Catal.*, 2022, 43, 139-147). As shown in Figure R1, the HER activity of Ru-WO_{3-x}/CP exhibits a sharp decrease in the deuterated solution compared to that in the non-deuterated solution. This result reveals an obvious contribution of proton transfer kinetics towards the HER process. Notably, this H/D isotopic effect is able to demonstrate that the proton transfer process is involved in the HER process, but it is still not possible to determine the rate limiting step of the reaction.

Figure R1. LSV curves of Ru-WO_{3-x}/CP in 1.0 M PBS prepared using D₂O and 1.0 M PBS prepared using H₂O.

11. The author said that "The second parallel circuit simulates the electrolyte-catalyst interfacial charge transfer, which is able to reflect the hydrogen intermediate adsorption behavior on the catalytically active sites." Please add some representative references after this conclusion.

Response: Thanks for the reviewer's suggestion. We have added some representative references, accordingly (Page 6, Line 47-48).

"The second parallel circuit simulates the electrolyte-catalyst interfacial charge transfer^{27, 53, 54},"

12. The authors highlight that the H atoms involved in the reaction tend to come from H transfer from H_xWO_{3-x} to Ru rather than water dissociation on Ru-WO_{3-x}/CP. The supply pathway of H on H_xWO_{3-x} should be given and explained in DFT calculation.

Response: Thanks for the reviewer's comments and suggestions.

The hydrogen in WO_{3-x} comes from two sources. The first one is from the protons in the electrolyte, which can be inserted into WO_{3-x} under negative potential. Another is from the dissociation of water on WO_{3-x} at large overpotentials. As revealed in Supplementary Figs. 22-24, the barrier for water dissociation on Ru is about 0.46 eV and that on WO_{3-x} is around 0.35 eV. The water dissociation on WO_{3-x} is more favorable than that on Ru.

For the Ru-WO_{3-x} catalytic system, protons can be directly inserted into WO_{3-x} to form H_yWO_{3-x}. The intercalation process of protons in WO_{3-x} can be written as:

This process occurs at a potential more negative to the peak of proton insertion into WO_{3-x}. And protons can be easily inserted into the hexagonal tunnels of WO_{3-x} and then coordinated with terminal oxygen atoms (*ACS Appl. Mater. Interfaces*, 2014, 6, 18901-18910). At the same time, the electrons injected from the electrode reduce the adjacent W atom from W⁶⁺ to W⁵⁺ (*ACS Appl. Mater. Interfaces*, 2016, 8, 13966-13972).

The proton insertion peak of Ru-WO_{3-x} in 1.0 M PBS is at 0.288 V vs. RHE (Fig. 4i), which is much more positive than the theoretical onset potential for proton adsorption over Ru for HER. This reveals that protons in the electrolyte can be

inserted into WO_{3-x} under the negative applied potentials. Then, the inserted protons in WO_{3-x} can spillover to Ru to facilitate the HER. As the further increase of overpotential, WO_{3-x} will dissociate the water to generate protons, which can also spillover to Ru.

Supplementary Figure 22. Energy barrier of water dissociation process on Ru of $\text{Ru-H}_x\text{WO}_{3-x}$ (a) and WO_{3-x} of $\text{Ru-H}_x\text{WO}_{3-x}$ (b). (TS represent the transition states).

Supplementary Figure 23. Side and top view illustrations of DFT models used for calculating the energy barrier of water dissociation process on Ru of $\text{Ru-H}_x\text{WO}_{3-x}$. (a) Initial state. (b) Transition state. (c) Final state. The blue, red and purple balls represent W, O and Ru atoms. The green and yellow balls represent the H atoms inserted into WO_{3-x} and the H atoms involved in HER.

Supplementary Figure 24. Side and top view illustrations of DFT models used for calculating the energy barrier of water dissociation process on WO_{3-x} of $\text{Ru-H}_x\text{WO}_{3-x}$.

(a) Initial state. (b) Transition state. (c) Final state. The blue, red and purple balls represent W, O and Ru atoms. The green and yellow balls represent the H atoms inserted into WO_{3-x} and the H atoms involved in HER.

Figure 4i. CV curve of Ru-WO_{3-x} recorded in 1.0 M PBS. Scan rate: 5 mV/s.

13. The authors indicate that protons were firstly inserted into the oxygen-deficient WO_{3-x} . Does these protons come from water or PBS electrolyte? In addition, in the microkinetic model, the adsorption state of OH^- should be considered.

Response: We thank the reviewer for raising the question. The protons are from the PBS electrolyte and we have elaborated this in our reply to your previous comment. In the microkinetic model, OH^- is a by-product after water dissociation and does not directly participate in the production of H_2 . Therefore, the adsorption of OH^- is not involved in the current microkinetic model.

14. Please carefully check this manuscript to revise some mistakes, such as "Fig. out in line 173", "29, 3.0 in line 220", "alkaline and neutral media water oxidation in line 274".

Response: Thanks for the reviewer's nice comment. We have carefully checked the entire manuscript and made the corresponding changes to the typos (**Page 6, Line 21**).

REVIEWER COMMENTS

Reviewer #2 (Remarks to the Author):

The revised version of the manuscript was significantly improved through the effort of the authors. The authors have answered the majority of the raised questions and concerns. Especially the calculation of the energy barriers of water dissociation on Ru and HWO_{3-x} helped. A difference in activation barrier of 0.1 eV, indeed could explain the 24 times higher rate (a rough calculation assuming the same rate constant and no other dependencies, results in a 70 times higher rate at room temperature). This appears convincing. Nevertheless, the authors should address the following points before the manuscript is ready for publication in Nature Communications.

1. Based on the suggested reaction pathways, per final H₂, two OH groups are produced. One during the water dissociation and one in the Heyrovsky step. What happens to the OH on the surface? A high concentration of surface OH could lead to the reverse reaction of OH+H to H₂O and be detrimental to the reaction. The authors should comment on the fate of the surface OH species.

2. Figure 5a: The authors calculated additional reaction steps (Water dissociation and recombination. They should now produce a figure 5a, which includes the full reaction path including their new calculations. The steps currently depicted in 5a are not critical, since they do not include the rate-limiting step.

3. The important question 9 from Reviewer 3, regarding the performance in alkaline medium has to be addressed in the manuscript. This is not reducing the importance of the manuscript but potentially reduces the number of resources spent by the community to perform such experiments. I suggest to the authors address this aspect towards the end, before the conclusion.

4. The scheme depicted in Figure 6 feels oversimplified and should not be part of the manuscript in the present form. Currently, it depicts how H present in the bulk of WO₃ recombines on Ru to form H₂. This is only a fraction of the HER reaction and, therefore, not informative for the reader. The scheme should depict the full reaction cycle, starting from H₂O, showing also the fate of OH. Furthermore, the authors could indicate all possible reaction pathways (only on WO₃ and only on Ru) and then distinguish what the preferred pathway is. In such a manner the figure would be informative and summarize the whole manuscript.

5. Legend in Figure 6: I suppose the labeling O and OV were swapped, otherwise the WO₃ would be extremely rich in vacancies.

Reviewer #3 (Remarks to the Author):

The authors did a fine job addressing my previous critiques. Since all my questions have been thoroughly answered, I suggest its acceptance and publication in Nature Communications.

Reviewer #2 (Remarks to the Author):

The revised version of the manuscript was significantly improved through the effort of the authors. The authors have answered the majority of the raised questions and concerns. Especially the calculation of the energy barriers of water dissociation on Ru and HWO_{3-x} helped. A difference in activation barrier of 0.1 eV, indeed could explain the 24 times higher rate (a rough calculation assuming the same rate constant and no other dependencies, results in a 70 times higher rate at room temperature). This appears convincing. Nevertheless, the authors should address the following points before the manuscript is ready for publication in Nature Communications.

Response: We appreciate the reviewer's constructive comments and suggestions, response to which shall greatly improve the quality of our manuscript. Below we give a point to point response to the questions/concerns raised by the reviewer.

1. Based on the suggested reaction pathways, per final H_2 , two OH groups are produced. One during the water dissociation and one in the Heyrovsky step. What happens to the OH on the surface? A high concentration of surface OH could lead to the reverse reaction of $\text{OH}+\text{H}$ to H_2O and be detrimental to the reaction. The authors should comment on the fate of the surface OH species.

Response: We thank the reviewer for the valuable suggestion. Accordingly, we have commented on the fate of the surface OH species in the revised manuscript (**Page 9, Lines 36-38**).

“The OH species on the surface of catalyst will undergo desorption and then be quickly captured by the buffer electrolyte. The OH concentration on the catalyst's surface is low because the buffer electrolyte can react quickly with the desorbed OH species to produce H_2O .”

2. Figure 5a: The authors calculated additional reaction steps (Water dissociation and recombination). They should now produce a figure 5a, which includes the full reaction path including their new calculations. The steps currently depicted in 5a are not critical, since they do not include the rate-limiting step.

Response: We thank the reviewer for the valuable comment and suggestion. Accordingly, we have updated Figure 5a as shown below.

Fig. 5a Calculated free energy diagram for HER on Ru- $\text{H}_x\text{WO}_{3-x}$ and $\text{H}_x\text{WO}_{3-x}$.

3. The important question 9 from Reviewer 3, regarding the performance in alkaline medium has to be addressed in the manuscript. This is not reducing the importance of the manuscript but potentially reduces the number of resources spent by the community to perform such experiments. I suggest to the authors address this aspect towards the end, before the conclusion.

Response: We thank the reviewer for the nice suggestion. Accordingly, we have added the following statement before the conclusion (**Page 5, Lines 13-15**).

“ WO_3 or WO_{3-x} gradually dissolves in alkaline electrolyte^{48,49}. Thus, it is not appropriate to apply Ru- WO_{3-x} in alkaline condition because of the stability issue.”

4. The scheme depicted in Figure 6 feels oversimplified and should not be part of the manuscript in the present form. Currently, it depicts how H present in the bulk of WO_3 recombines on Ru to form H_2 . This is only a fraction of the HER reaction and, therefore, not informative for the reader. The scheme should depict the full reaction cycle, starting from H_2O , showing also the fate of OH. Furthermore, the authors could indicate all possible reaction pathways (only on WO_3 and only on Ru) and then distinguish what the preferred pathway is. In such a manner the figure would be informative and summarize the whole manuscript.

Response: Thanks for the reviewer’s comment. Accordingly, we have updated Figure 6 and depicted the full reaction cycle. All possible reaction pathways are provided in the revised Figure 6 with the preferred pathway indicated.

Fig. 6 Illustration of hydrogen spillover. Schematic diagram showing how hydrogen spillover from WO_{3-x} to Ru enhances HER in neutral environment. The white and red arrows indicate the formation process of OH and H from dissociation of water, respectively. Black arrows indicate the transfer process of H. The purple arrows indicate the desorption of OH adsorbed on the surface. The blue, yellow and green arrows indicate the different ways of formation of molecular hydrogen.

5. Legend in Figure 6: I suppose the labeling O and O_v were swapped, otherwise the WO_3 would be extremely rich in vacancies.

Response: Thanks for the reviewer’s comment. We have revised this legend in Figure 6 accordingly.

Reviewer #3 (Remarks to the Author):

The authors did a fine job addressing my previous critiques. Since all my questions have been thoroughly answered, I suggest its acceptance and publication in Nature Communications.

Response: We are grateful to the reviewer's efforts on evaluating our manuscript, and we are excited about the reviewer's recognition on this work. We truly appreciate the constructive comments from the reviewer to improve the quality of this work.

REVIEWERS' COMMENTS

Reviewer #2 (Remarks to the Author):

The authors answered all remaining questions thoughtfully. I recommend the manuscript for publication in Nature Comm.

Reviewer #2 (Remarks to the Author):

The authors answered all remaining questions thoughtfully. I recommend the manuscript for publication in Nature Comm.

Response: We are grateful to the reviewer's efforts to evaluate our manuscript and are excited about the reviewer's recognition on this work. We truly appreciate the constructive comments from the reviewer to improve the quality of this work.